# Reproducing extracellular matrix adverse remodelling of non-ST myocardial infarction in a large animal model

Paolo Contessotto [1,2,14], Renza Spelat [1,14], Federico Ferro [1], Vaidas Vysockas [3], Aušra Krivickienė [4,5], Chunsheng Jin [6], Sandrine Chantepie[7], Clizia Chinello[8], Audrys G. Pauza [9,10], Camilla Valente [2], Mindaugas Rackauskas [11], Alvise Casara [12], Vilma Zigmantaitė [3], Fulvio Magni [8], Dulce Papy-Garcia[7], Niclas G. Karlsson [6,13], Eglė Ereminienė [4,5], Abhay Pandit[1] ✉ & Mark Da Costa [1] ✉

The rising incidence of non-ST-segment elevation myocardial infarction (NSTEMI) and associated long-term high mortality constitutes an urgent clinical issue. Unfortunately, the study of possible interventions to treat this pathology lacks a reproducible pre-clinical model. Indeed, currently adopted small and large animal models of MI mimic only full-thickness, ST-segment-elevation (STEMI) infarcts, and hence cater only for an investigation into therapeutics and interventions directed at this subset of MI. Thus, we develop an ovine model of NSTEMI by ligating the myocardial muscle at precise intervals parallel to the left anterior descending coronary artery. Upon histological and functional investigation to validate the proposed model and comparison with STEMI full ligation model, RNA-seq and proteomics show the distinctive features of post-NSTEMI tissue remodelling. Transcriptome and proteome-derived pathway analyses at acute (7 days) and late (28 days) post-NSTEMI pinpoint specific alterations in cardiac post-ischaemic extracellular matrix. Together with the rise of well-known markers of inflammation and fibrosis, NSTEMI ischaemic regions show distinctive patterns of complex galactosylated and sialylated N-glycans in cellular membranes and extracellular matrix. Identifying such changes in molecular moieties accessible to infusible and intra-myocardial injectable drugs sheds light on developing targeted pharmacological solutions to contrast adverse fibrotic remodelling.

Myocardial infarction (MI) is an acute complication of coronary artery disease and it is the leading cause of cardiovascular-related worldwide mortality[1]. In addition, COVID-19 infection was recently shown to be an additional major risk factor in non-hospitalised cases[2]. Patients who survive an MI have profound cardiac tissue alterations that can lead to lethal heart failure later on[3]. A marked rise in non-ST-elevation myocardial infarctions (NSTEMIs) in hospitalised cases has emerged over the last two decades[4,5]. Moreover, despite the smaller ventricular wall areas affected by the ischaemic event compared to ST-elevation myocardial infactions (STEMIs), registry data show that NSTEMIs are associated with long-term mortality higher than STEMIs[6–9].

In current preclinical studies, MI is mainly reproduced as STEMI in rodents and large animals (porcine and ovine) by the ligation of the coronary arteries arising from the left coronary artery, specifically with

---

an established preference for the ligation of the left anterior descending coronary artery (LAD)[10,11]. Ligation of the LAD results in an extended infarct in the left ventricle, associated with high experimental mortality and a poor reflection of most hospitalised clinical cases. Indeed, most MIs currently reported in the clinics are either partial or non-transmural infarcts, often involving multiple small regions of the left ventricle[5,12]. Only a few studies have paved the way by optimising the extension of the infarct up to approximately 25% of the infarct's left ventricular mass and bringing the overall experimental mortality down to around 17%[13,14].

Therefore, there is a need in the field to adopt clinically relevant models to study NSTEMI pathophysiology and also reveal its functional differences with STEMI induction.

In all the previously adopted preclinical models of MI, collagen deposition in the myocardial left ventricular wall (cardiac fibrosis) is a hallmark of non-lethal cardiac ischemic events. Fibrosis compensates, albeit poorly, for the extensive loss of cardiomyocytes[15]. Indeed, the entire post-infarction process begins with a sterile immunological response involving different populations of macrophages and inflammatory cells that have recently been characterised by single-cell RNA sequencing[16,17]. Notably, post-ischaemic myocardial remodelling disrupts the initial balance of glycosaminoglycans (GAGs), proteoglycans and glycans, which are present in the extracellular matrix (ECM) and cell membrane of cardiac cell populations (cardiomyocytes, fibroblasts, endothelial cells)[18,19]. Consequently, adverse remodelling affects the structural and mechanical stability of the ECM environment. This leads to further imbalances in molecular pathways that include the recruitment of growth factors such as vascular endothelial growth factor (VEGF), platelet-derived growth factor (PDGF) and fibroblast growth factor (FGF). Large animals, specifically sheep, have been extensively used to evaluate the recovery of heart functionality following STEMI because of the similarity in organ volume to humans[20,21]. Therefore, in this study, we first validated the functional differences between NSTEMI and STEMI within an ovine model and further analysed the distinctive molecular features of the NSTEMI model at both an acute (7 days) and a late (28 days) time point. Specifically, we have studied the ischaemic, border and remote regions at the different time points post-NSTEMI by histology, transcriptomics, proteomics and glycomics. Indeed, given the key role of the modulation of post-ischaemic cardiac ECM in developing translational therapies and the lack of an established animal model that mimics partial thickness myocardial infarction, a clinically relevant model of NSTEMI was urgently needed in the cardiovascular field[22,23].

## Results

### Functional impairment in a model of NSTEMI compared to STEMI

Current preclinical models of full-thickness infarcts (STEMIs) are based on the ligation of the LAD or variations of this procedure in the LAD territory[10,24]. A significant limitation of the LAD ligation model in its clinical resemblance is that proximal occlusion of the coronary is often fatal. Indeed, the incidence of NSTEMIs currently exceeds the actual clinical infarcts that LAD ligation-based models aim to reproduce[22,25,26]. In this study, multiple ligations were performed lateral and parallel to the LAD from the first diagonal up to 3–4 cm from the apex to induce multiple non-transmural infarcts in the left ventricle (Fig. 1a, b and Supplementary Fig. 1). The proposed model of NSTEMI was performed in a cohort of 21 sheep and compared with seven additional sheep subjected to full-occlusion of the first LAD diagonal branch. Four sheep died during the NSTEMI procedure and two as the immediate consequence of STEMI, resulting in an overall mortality of 19.04% and 28.57%, respectively. In the NSTEMI group vagus nerve stimulation during intubation caused two deaths, one was due to left ventricle rupture and one of uncertain cause at the premedication stage. Thus, only one sheep died from a complication of NSTEMI. Six sheep were

sacrificed on day 7 (d7) and the remaining eleven on day 28 (d28) post-NSTEMI as the final endpoint to evaluate both functional and histological alterations.

The current NSTEMI model causes focal infarcts, associated with a limited yet significant reduction in EF ($8.52 \pm 7.88\%$, $P = 0.03$) on d7 post-surgery (Fig. 1c). Three weeks later (d28), EF decreased by $10 \pm 8.31\%$ ($P = 0.009$) relative to the pre-MI levels (Fig. 1c). On the contrary, as well-known in the field, full-thickness infarcts induce marked drops in ejection fraction (EF)[27,28]. Therefore, to validate the current model, we compared it with a standard full-occlusion ligation of the first diagonal branch which caused full-thickness STEMIs in sheep of same age and gender. By comparing NSTEMI with STEMI, in the latter we noticed a greater reduction in EF both at d7 ($15.4 \pm 4.4\%$, $P = 0.03$) and d28 ($14.8 \pm 3.3\%$, $P = 0.03$) post-ligation (Supplementary Fig. 2a), together with a marked rise ($P < 0.05$) in troponin I level from d1 to d3 post-surgery (Supplementary Fig. 2b). Haemodynamic parameters showed no difference in stroke volume and cardiac output (Supplementary Fig. 2c), conversely ECGs showed typical changes of STEMI (Supplementary Fig. 2d).

Post-ischaemic remodelling involves different degrees of dilatation, hypertrophy and collagen scarring. This process occurs over weeks and months, and it is influenced by multiple factors, including the size and site of the infarct, whether the infarct is transmural (STEMI) or not (NSTEMI), the amount of stunning of the peri-infarct myocardium, the patency of the related coronary artery and local trophic factors[29,30]. Since NSTEMI is not the result of complete occlusion of a coronary artery, it usually affects a small area or patchy areas of the ventricular muscle rather than the entire thickness of the local ventricular wall. Indeed, as expected, given the nature of this type of MI, the presented model did not significantly vary in left ventricular end diastolic and systolic diameters (EDD and ESD) (Fig. 1c). Therefore, to further evaluate the functional impairment seen in the proposed NSTEMI model, we analysed the loss of cardiac contractility in the specific left ventricular segments affected by NSTEMI through regional wall-motion index (WMI) analysis (Fig. 1d). On d7 post-NSTEMI wall-motion impairments were significant in basal anterior/anteroseptum, (WMI = $1.73 \pm 0.56$ and $1.82 \pm 0.6$, $P < 0.001$), mid anteroseptum/anterior (WMI = $1.82 \pm 0.6$ and $2.14 \pm 0.45$, $P < 0.001$), apical anterior (WMI = $1.64 \pm 0.64$, $P < 0.01$), and apical septum (WMI = $1.41 \pm 0.58$, $P < 0.05$) segments. This widespread wall-motion deficit persisted in all the affected segments on d28 post-NSTEMI (Fig. 1d).

### Ischaemic damage and consequent adverse remodelling following NSTEMI

Importantly, we have confirmed the described NSTEMI-induction approach with clinical data since electrocardiograms (ECGs) post-ligation highlighted comparable changes in T wave inversion in leads I, II, III and aVF (Fig. 1e and Supplementary Fig. 3). In line with the current findings on left ventricular EDD and ESD, the reduction in fractional shortening (FS) on d7 and d28 post-surgery was not significant (Supplementary Fig. 4a). Therefore, echocardiography, troponin I and ECG data followed the clinical criteria which concur to define the current model as representative of an NSTEMI event[31].

The non-transmural nature of the induced NSTEMI infarcts became apparent with explantation on d28 post-MI (Fig. 1b). During the surgical procedure, the localisation of the infarct was defined by its proximity to the blue suture used to perform the multiple ligations (Supplementary Fig. 1b–f). Indeed, once the hearts were sliced at a thickness of 1 cm, the NSTEMI regions were detectable by the discolouration of the left ventricular wall (Fig. 1b). Therefore, sampling was carried out from the ischaemic core progressively to the border and remote regions (Supplementary Fig. 4b). Specimens were isotropically uniformly oriented to avoid any bias when evaluating cardiomyocytes and vasculature structures[32].

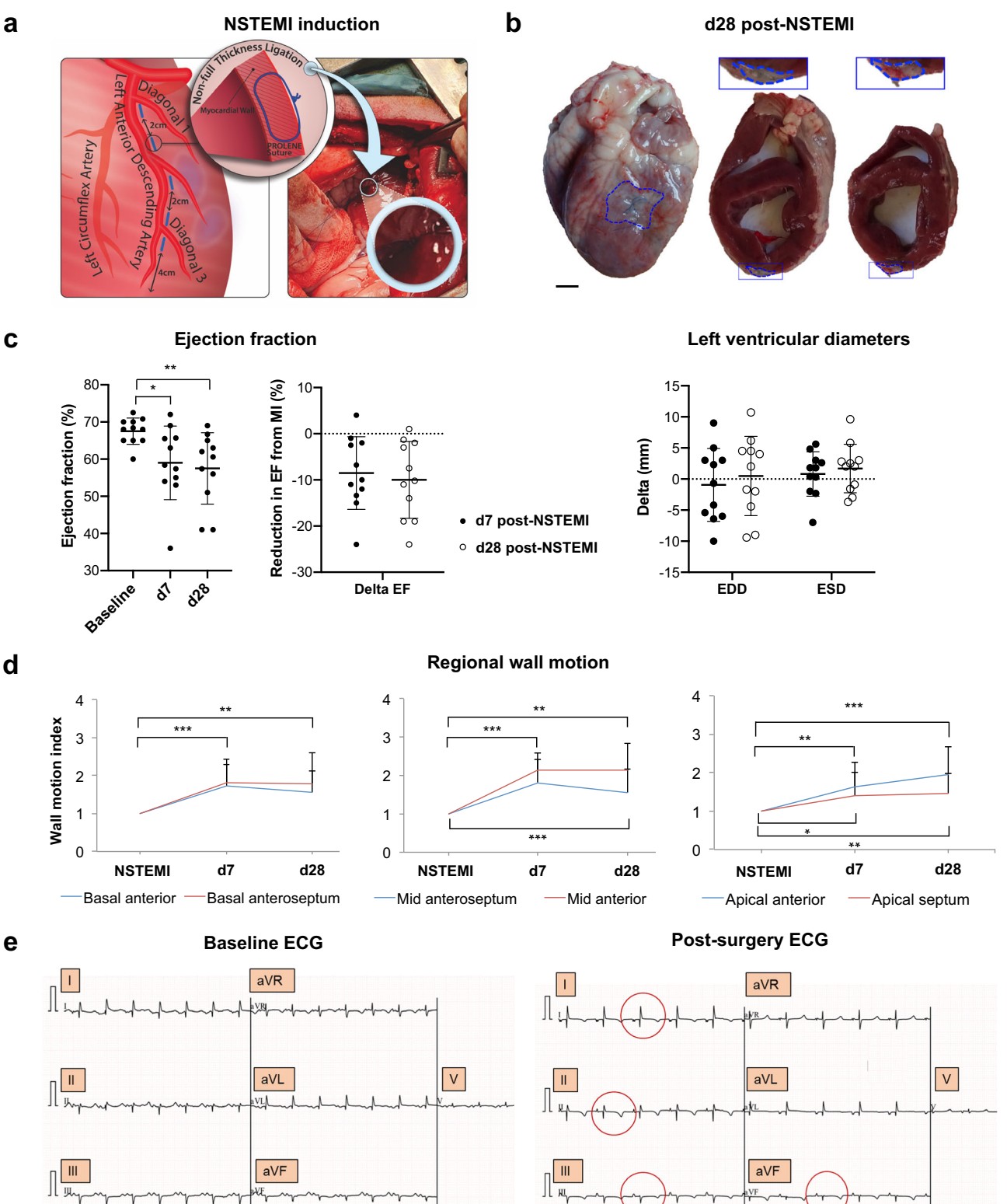

It is well known that the necrotic phase responsible for the loss of cardiomyocytes in the left ventricle and the consequent highly pro-inflammatory microenvironment represents the initial immediate steps post-ischaemia[33]. Moreover, a valid model of MI also needs to reproduce the long-term effects on the border zone region indirectly affected by the complete tissue disruption in the ischaemic core[34,35]. This study detected a progressive vacuolisation inside the cardiomyocytes from d7 to d28 after NSTEMI induction (Fig. 2a). Disruption of the intercalated discs and lipid droplet accumulation

(Fig. 2a and Supplementary Fig. 4c) were seen exclusively in the acute phase (d7). In contrast, fibrotic deposition fully developed at the endpoint (d28) of the study (Fig. 2a,b and Supplementary Fig. 4c). Specifically, intercalated disk disruption highlighted the structural disassembly of the cardiomyocyte myofibril apparatus needed for its cellular contraction and thus the synchronous beating of the whole organ. In addition, the accumulation of dense bodies in the mitochondria of these cardiomyocytes further supported the evidence of ischaemic injury (Fig. 2c). Together, these data show

**Fig. 1 | Clinically relevant ovine model of NSTEMI. a** Schematics of the multiple ligation procedure to induce NSTEMI infarcts. **b** Representative photographs of 8-month-old explanted and axially-cut sheep hearts 28 days post-NSTEMI. Blue Prolene® sutures were used to track NSTEMI infarcts (framed in blue). $n = 11$ animals. Insets shown at higher magnification above. Scale bar, 1 cm. **c** Left, ejection fraction (EF) absolute values before ligation (baseline), 7 (d7) and 28 (d28) days post-NSTEMI and relative decrease in EF on d7 and d28 post- surgery (left). Right, measurement of left ventricular end diastolic diameter (EDD) and systolic diameter (ESD) on d7 and 28 post-NSTEMI. $n = 11$ animals. **d** Regional wall-motion analysis in the main six cardiac segments affected by the induction of NSTEMI. $n = 11$ animals. **e** Representative electrocardiogram (ECG) before NSTEMI-induction (left) and

post-surgery (right). Changes in T wave inversion, in leads I, II, III and aVF are circled in red. $n = 4$ animals. Kruskal-Wallis with Dunn's multiple comparisons test ($P = 0.034$ for baseline vs. d7 and $P = 0.009$ baseline vs. d28) in (**c**, left), two-tailed Wilcoxon test in (**c**, right), and unpaired t-test with two-stage step-up Benjamini method (basal anterior $P = 0.0003$ baseline vs. d7 and $P = 0.004$ vs. d28, basal anteroseptum $P = 0.0002$ baseline vs. d7 and $P = 0.005$ vs. d28, mid anteroseptum $P = 0.0002$ baseline vs. d7 and $P = 0.007$ vs. d28, mid anterior $P < 0.0001$ baseline vs. d7 and $P < 0.0001$ vs. d28, apical anterior $P = 0.003$ baseline vs. d7 and $P = 0.0003$ vs. d28, apical septum $P = 0.03$ baseline vs. d7 and $P = 0.009$ vs. d28 in (**d**). *$P < 0.05$, **$P < 0.01$, ***$P < 0.001$. Data are plotted showing the mean and standard deviation in (**c**, **d**). Source data are provided as a Source Data file.

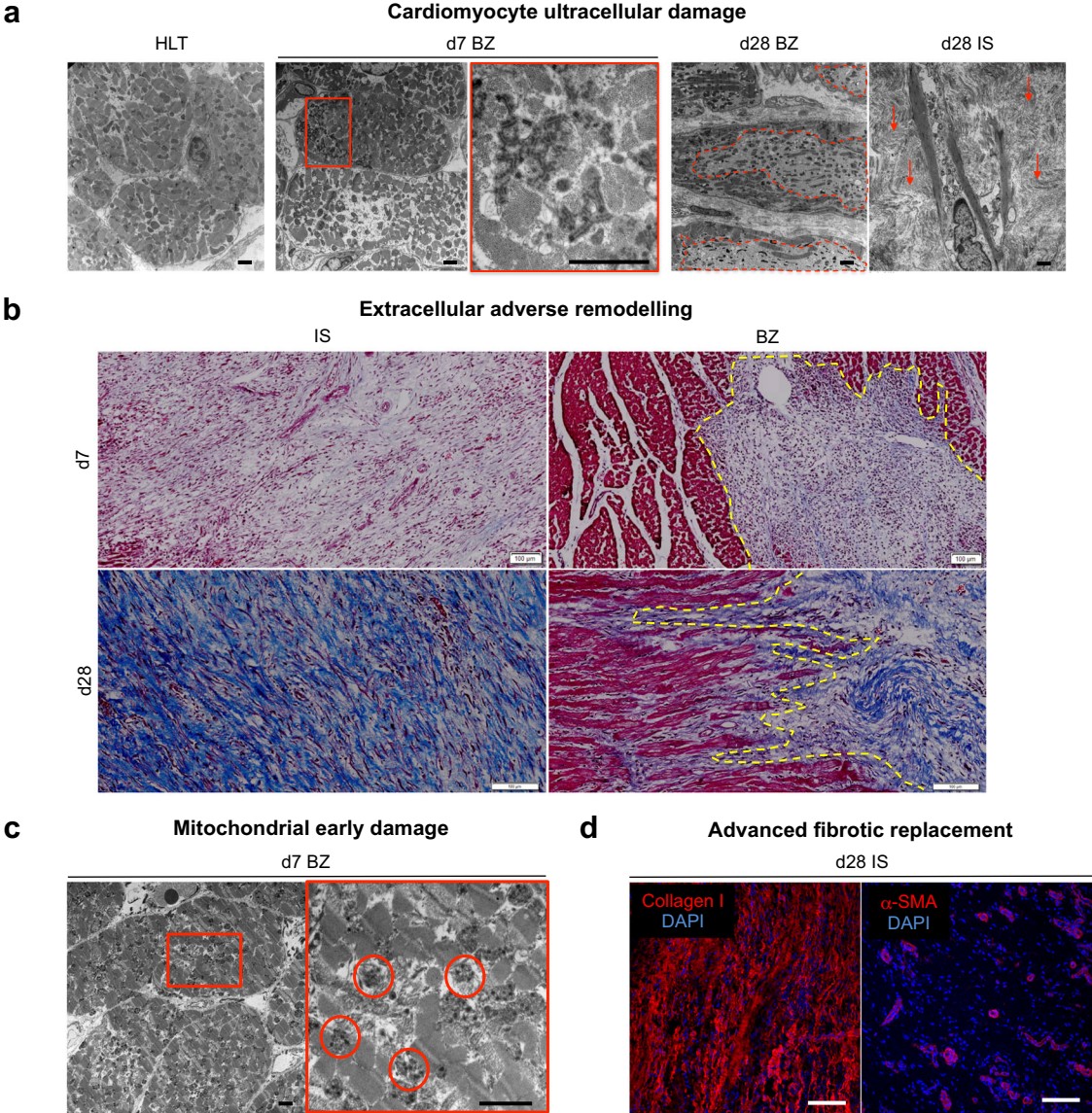

**Fig. 2 | Ischaemic cellular damage and extracellular adverse remodelling following NSTEMI. a** Representative TEM micrographs showing ultracellular damage from healthy (HLT, far left) to border zone (BZ) cardiomyocytes starting from intercalated disks disruption on d7 post-NSTEMI (centre, with inset), to extended vacuolisation (dashed red line) on d28 post-NSTEMI (right), surrounded by collagen deposition (arrows) by myofibroblasts in ischaemic core (IS). $n = 5$ HLT and d7, $n = 7$ d28 animals. Scale bars, 2 μm. **b** Representative Masson's Trichrome

staining of IS and BZ regions of NSTEMI-infarcted tissues on d7 and d28. $n = 5$ animals per group. Scale bars, 100 μm. **c** Representative TEM micrographs of mitochondria in cardiomyocytes located in the BZ. Inset shows accumulation of dense bodies (circled in red) on d7 post-NSTEMI. $n = 5$ animals. Scale bars, 2 μm. **d** Confocal microscopy images indicating fibrotic replacement (left) and sparse α-SMA$^+$ arterioles (right) in IS on d28. $n = 5$ animals. Scale bars, 20 μm.

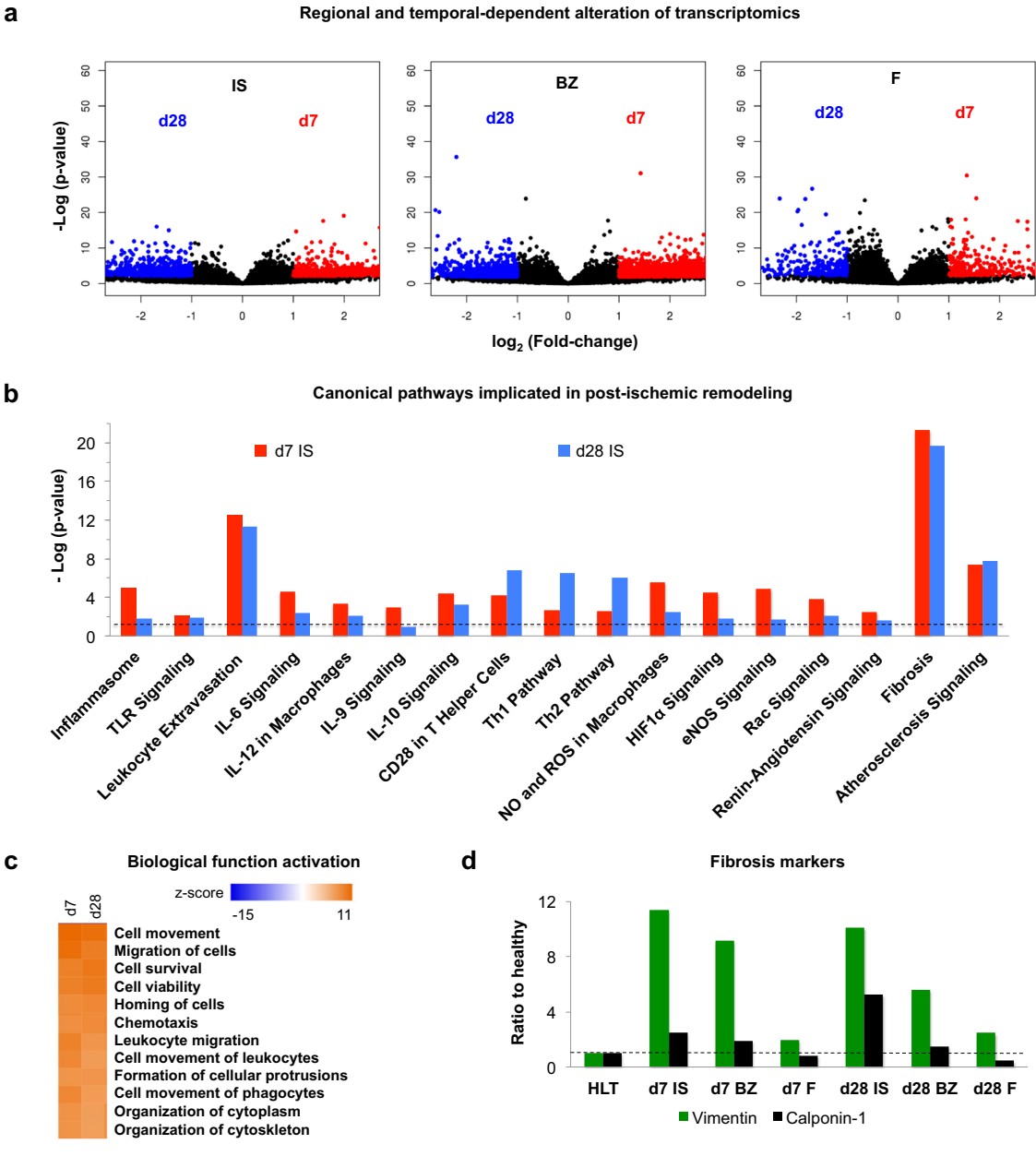

**Fig. 3 | Post-ischaemic pathways alteration following NSTEMI. a** Volcano plots showing the total genes significantly upregulated across the core ischaemic (IS), border (BZ) and remote (F) regions sampled on d7 and d28 post-NSTEMI. $n = 4$ animals per group. **b** Canonical Pathways Analysis (IPA®) on differentially expressed genes (DEG) from RNA-seq data. Cut-offs of log2(fold-change) > 1.5 and log2(fold-change) <−1.5, and adjusted-$P$ < 0.05 were set. Dashed line shows a threshold of -Log($p$-value) of 1.3, corresponding to $P = 0.05$. All DEG data were normalised to healthy baseline left ventricular samples. $n = 4$ animals per group. **c** Main activated biological functions listed by highest z-score from IPA® analysis on DEG. $n = 4$ animals per group. **d** Relative abundance of myofibroblast-related markers vimentin and calponin-1 as detected by nLC-ESI-MS/MS analysis on IS samples. Each analysed sample was a pool of samples coming from three animals. Source data are provided as a Source Data file.

that the current model of NSTEMI reproduces the sequential inflammatory and fibrotic remodelling of the infarcted region, resulting in a non-full-thickness scar formation, which is characteristic of the clinical cases of NSTEMI.

Histology showed the extended loss of cardiomyocytes from the ischaemic core to the border zone on d7 post-NSTEMI induction (Fig. 2b). From the end of the first week after MI and through the following weeks the myofibroblasts secrete collagen[36], and this event was also observed in the current model (Fig. 2b, d). Moreover, an irregular vascularisation in the fibrotic areas was detected by immunostaining for alpha-smooth muscle actin-positive (α-SMA+) arterioles. (Fig. 2d).

## Transcriptome and proteome of NSTEMI infarcts

To investigate the molecular profiling of the current model of NSTEMI, we analysed the transcriptome and proteome of the ischaemic core, border, and remote regions. Bulk RNA-seq analysis (adjusted-$P$ of 0.05, mean quality score 38.41) highlighted pools of differentially expressed genes (DEG) at each time point post-ligation, depending on the distance from the ischaemic core (Fig. 3a). A clear regional- and temporal-dependent transcriptome alteration was seen in the ischaemic core, border and remote regions (Fig. 3a, Supplementary Fig. 5, and Supplementary Data 1–5). Specifically, 1079 transcripts were upregulated [log2(fold-change) > 1.5] and 805 downregulated [log2(fold-change) <−1.5] in the ischaemic core on d7 compared to d28 post-NSTEMI

(Supplementary Data 1). Moreover, 1855 transcripts were upregulated and 667 downregulated in the border zone; finally, in the remote region, 284 were increased and 233 decreased on d7 compared to d28 post-surgery (Supplementary Fig. 5, and Supplementary Data 2–3).

DEG data derived from infarcted tissues on d7 and d28 vs healthy samples were analysed by Ingenuity Pathway Analysis (IPA®) to identify which canonical pathways are significantly altered following NSTEMI. Many distinctive pathways associated with myocardial ischaemic pathophysiology were detected, such as Fibrosis, Atherosclerosis and Renin-angiotensin signalling both on d7 and 28 post-surgery (Fig. 3b and Supplementary Data 6). In addition, several pathways linked to the inflammatory response emerged in the d7 post-NSTEMI group, including Hypoxia-inducible Factor (HIF1α), Endothelial NOS (eNOS), and Interleukin-6, −9, −10, −12 (IL), Leucocyte extravasation and Inflammasome signalling (Fig. 3b). On d7 and d28 post-NSTEMI biological functions linked to inflammatory cell recruitment ranked as the top activated ones (z-score above 10) (Fig. 3c and Supplementary Data 7). Indeed, Causal Network Analysis (IPA®) on DEG associated with higher HIF1α expression predicted the activation of inflammatory markers such as IL-1β, IL-6, and tumour necrosis factor (TNF) on d7 (Supplementary Fig. 6a and Supplementary Data 8). HIF1α was also associated with the predicted activation of fibrotic molecular profiling such as transforming growth factor-β (TGF-β), matrix metalloproteinase-2 and −9 (MMP-2, −9) and lysyl oxidase (LOX) on d28 in the ischaemic core (Supplementary Fig. 6b and Supplementary Data 9).

Proteomic analysis was run to further validate the molecular changes seen by RNA-seq and pathway analysis showing a post-ischaemic remodelling response. In line with the RNA-seq outcome, Upstream Regulator Analysis (IPA®) on proteomic data showed significant activation of IL-1β, TNF, IL-6, TGF-β, IFN-γ in the ischaemic core 28 days post-NSTEMI (Supplementary Data 10). In addition, nLC-ESI-MS/MS data from total protein extracts of ischaemic core, border and remote region (false discovery rate below 1%) highlighted well-known markers of fibrotic replacement (Fig. 3d and Supplementary Data 11). Interestingly, a progressively increased calponin-1 and vimentin abundance was seen from the remote to the ischaemic core regions both on d7 and d28 post-NSTEMI (Fig. 3d). Specifically, when compared with the healthy left ventricular myocardial sample, vimentin ratio increased to 9.13-fold and 11.37-fold on d7 and to 5.58-fold and 10.10-fold on d28 in the border and ischaemic core regions, respectively. Also, the calponin-1 ratio increased to 1.87-fold and 2.5-fold on d7 and to 1.5-fold and 5.28-fold on d28 in the border and ischaemic core regions, respectively (Fig. 3d and Supplementary Data 11). Importantly, gene-annotation enrichment analysis using Database for Annotation, Visualization, and Integrated Discovery (DAVID) software on RNA-seq data highlighted distinctive glycan alterations in the post-ischaemic remodelling in the current NSTEMI model (Supplementary Fig. 7 and Supplementary Data 12-15). Given the outcome of RNA-seq and proteomic pathway analyses, we noticed a key involvement of glycan moieties, since both on d7 N-glycan biosynthesis (Supplementary Fig. 7a and Supplementary Data 12) and on d28 GAGs biosynthesis (Supplementary Fig. 7b and Supplementary Data 15) emerged among the biological functions of Kyoto Encyclopedia of Genes and Genomes (KEGG) pathway database with the highest enrichment score. As the importance of glycoproteins in the pathophysiology of MI has only recently emerged[18,37], we performed advanced glycomics on N-linked glycans extracted from the cellular membrane and ECM proteins in ischaemic, border and remote regions.

## Distinct glycoprofile in the ischaemic tissue following NSTEMI
Considering the findings highlighted by the pathway and functional annotation analyses on RNA-seq data, we scrutinised the altered glycan composition in the cellular membrane and ECM proteins of infarcted NSTEMI tissue. Here, we dissected the glycome under the inflammatory (d7) and fibrotic (d28) conditions following NSTEMI by advanced glycomic analysis of the N-glycans expressed in the left ventricular membrane and ECM protein fraction. To achieve this, during the sample processing N-linked glycans were released and analysed by LC-MS. Following annotation procedures, 103 putative N-glycan structures were identified, including 10 high mannose, 15 hybrid and 78 complex-type glycans (Fig. 4 and Supplementary Data 16). Since the relative abundance of these structures varied across healthy and infarcted tissues, hierarchical clustering analysis was performed for high-mannose and complex N-linked glycans (Fig. 4a–e). High-mannose N-linked glycans clustering reflected the main distinctions among ischaemic core (IS), border (BZ) and remote regions (F) (Fig. 4a). The subgroup consisting of high mannose N-glycans in the healthy myocardium and F regions showed a global similarity of only 38% to IS on d7 and d28 high mannose N-glycans (based on distribution found by MS) (Fig. 4a). This is in contrast to the post-NSTEMI (d7 and d28), where IS regions showed a global 76% similarity in high mannose N-glycans when grouped. The difference in the level of high mannose glycans showed distinct features at d7 and d28 post-NSTEMI depending on the distance from the IS region (Fig. 4b). In IS, high mannose structures decreased to 21-24% compared to healthy (HLT) tissue (32%) both on d7 and d28. At the same time, the remote (F) area appeared not to be affected based on expression compared to healthy. The border zone (BZ) also appeared to be affected by a decrease in high mannose on d7 (23%), but recovered after 28 days, to 38%, a level similar to those of healthy (32%) and less affected remote areas (F) (32-34%) (Fig. 4b). The opposite effect, consistent with the trends of the high mannose structure, was seen in complex N-glycans (Fig. 4c,d). Indeed, the level of complex N-glycans detected in IS regions on d7 and d28 had a limited similarity (39%) with healthy myocardium (Fig. 4c). Ischaemic core regions clustered together both on d7 and d28, and hence showed no signs to recover due to time, with a low (48%) similarity with the less affected remote (F) ones (Fig. 4c). Starting from d7, both in IS and BZ the level of complex glycans was increased (71%) compared to those of healthy and less affected F areas (55%) (Fig. 4d), until d28 in IS (75%). Finally, in IS regions hybrid N-glycan expression decreased to 4–5% from the initial 13% in healthy (Fig. 4e).

Therefore, we further investigated which specific glycan structures were involved in the observed altered molecular profile. Within complex N-glycans, we noticed a clear trend towards a progressive increase in sialylated structures (Fig. 4f and Supplementary Data 16). Sialylation is a well-known glycosylation type present across cardiac cellular populations such as cardiomyocytes and endothelial cells[18,38]. In particular, in the ischaemic tissues we observed an increased expression of different sialic acid forms such as neuraminic acid (NeuAc) and N-glycolylneuraminic acid (NeuGc) compared to healthy cardiac ventricular tissue. Specifically, these two forms of sialic acid followed an opposite pattern of expression since NeuGc - which is a well-known xeno-antigen in humans - peaked at d7 post-NSTEMI (from 34% to 44% more than healthy) before dropping to baseline at d28 (Supplementary Fig. 8a). On the contrary, NeuAc increased from d7 to d28 post-NSTEMI in all infarcted regions, including F, reaching a maximum rise of over 40% in IS on d28 (Supplementary Fig. 8a). By analysing sialic acid linkage-type, we noticed a pattern of peaked increase in α-(2,6)-sialic acid linkage-type at d7 in ischaemic regions, and thus within the inflammatory phase, in contrast with a progressive increase of α-(2,3)-sialylation in all regions - infarcted and not - over the remodelling from d7 to d28 (Supplementary Fig. 8b). These findings led to the evaluation of possible association with sialic acid-binding ligands known to be expressed by the infiltrating immune cell populations, such as monocytes and macrophages[39–42]. Indeed, we have observed that both sialic acid-binding immunoglobulin-type lectins −1 and −15 (Siglec-1 and −15) - which are highly present in monocytes and macrophages[40,41] - were markedly overexpressed ($P < 0.05$) at d7 and significantly dropped 21 days later, in line with a resolved inflammation

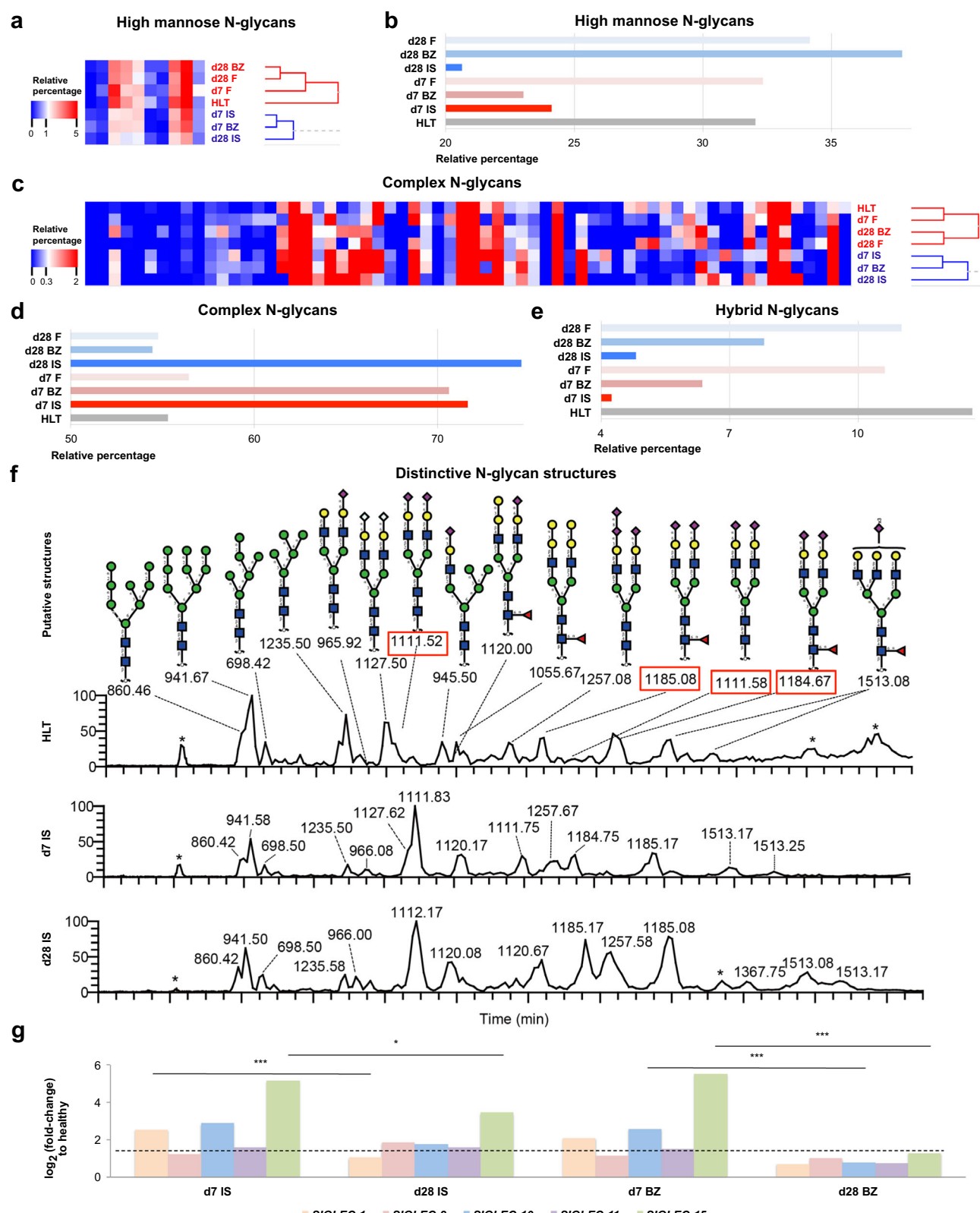

**a** High mannose N-glycans

**b** High mannose N-glycans

**c** Complex N-glycans

**d** Complex N-glycans

**e** Hybrid N-glycans

**f** Distinctive N-glycan structures

**g**

In addition, *SIGLEC*−15 expression was significantly increased both in the ischaemic core ($P < 0.05$) and border zone ($P < 0.001$) region, and *SIGLEC*−10, another marker of infiltrating activated monocytes and macrophages[42], decreased ($P < 0.001$) from d7 to d28 (Fig. 4g). To highlight the similarity of the proposed model to the clinical cases of NSTEMI, we report (Fig. 4f and Supplementary Data 16) a group of highly present complex sialylated N-glycan structures (*m/z*

1111.52 (NeuAc$_2$Hex$_5$HexNAc$_4$), 1185.08 (NeuAc$_2$Hex$_5$HexNAc$_4$dHex$_1$) that are shared with recently characterised ones in MI patients[43].

An additional distinct pattern was seen in the expression of terminal α-galactose (α-gal), which in humans is a well-known xeno-antigen. Indeed, a striking increase (over 30%) in α-gal on d7 post-NSTEMI (in IS and BZ regions) was completely lost on d28 in both regions (Supplementary Fig. 8c). We also analysed released O-linked

**Fig. 4 | Distinct glycoprofile in the infarcted heart following NSTEMI. a** Ten high mannose N-glycans putative structures detected by LC-ESI-MS/MS and analysed by hierarchical clustering. **b** Relative percentage of high mannose among total N-glycans putative structures across healthy (HLT), core ischaemic (IS), border (BZ) and remote (F) region myocardial cellular membrane samples on d7 and d28 post-NSTEMI. **c**, 78 complex N-glycans putative structures detected by LC-ESI-MS/MS and analysed by hierarchical clustering. **d**, **e** Relative percentage of complex (**d**) and hybrid (**e**) among total N-glycans putative structures across HLT and infarcted myocardial cellular membrane samples on d7 and d28 post-NSTEMI. **f** Extracted ion chromatography (EIC) showing the most abundant N-linked glycans in the membrane protein extracts from HLT myocardium and IS on d7 and d28 post-NSTEMI.

$m/z$ values framed in red indicate N-glycan structures, which were found to increase also in MI patients' sera by Lim et al.[43]. **g** Gene expression levels of *SIGLEC*−1, −2, −10, −11 and −15 at d7 and 28 post-NSTEMI from RNA-seq data on IS and BZ samples. $n = 4$ animals per group. Regions of infarcted hearts are labelled as follows: IS = ischaemic core, BZ = border, F = remote zone from the infarct. Data are representative of two independent experiments. In (**a**–**f**), each analysed sample was a pool of samples collected from three individuals per group and region and investigated by LC-ESI-MS/MS. In (**g**), RNA-seq data analysis was run using DESeq2 (dashed line indicates the threshold set to $\log_2(\text{fold-change}) > 1.5$), and significant differences were assessed by two-sided Wald's test using Benjamini−Hochberg method. *$P < 0.05$, ***$P < 0.001$. Source data are provided as a Source Data file.

glycans in NSTEMI, identifying the appearance of post-NSTEMI sialylated structures, mainly present during the early (d7) phase of remodelling (Supplementary Fig. 8d and Supplementary Data 16).

Altogether, these data indicated a distinct glycoprofile in the NSTEMI cardiac tissue characterised by a higher abundance of complex N-glycans and in particular NeuAc (both (α-(2,3)- and α-(2,6)-sialic acid linkage) on d7 as well as on d28 post-NSTEMI (Fig. 4f). However, a marked increase in NeuGc and α-gal was associated only with the early phase of remodelling (d7), but lost through the endpoint (d28) when NeuAc – and in particular α-(2,3)-sialylation – were highly present.

### An irreversibly altered extracellular matrix shows specific changes in the HS sulfation pattern

To further investigate the extensive changes in the ECM following the induction of NSTEMI infarcts, we performed additional analyses on essential components of the myocardial ECM, such as GAGs. Indeed, the restructuring of the ECM is one of the main consequences of post-ischaemic remodelling in the left ventricular wall[15,33,36]. Specifically, GAGs can regulate inflammation and angiogenesis, influencing the remodelling response[44,45]. The ischaemic core region was initially screened for sulfated GAGs (sGAGs) by Alcian Blue staining. This confirmed their distribution within the fibrotic regions compared with Masson's Trichrome staining (Fig. 5a). Then, sGAGs were extracted from samples of the infarct zone harvested on d7 and d28 post-NSTEMI. After total GAGs quantification, the ratio between heparan sulphate (HS) and chondroitin sulfate (CS) was calculated by a subtraction method after enzymatic digestion of the samples with chondroitinase ABC. An increase in CS was seen post-NSTEMI, bringing the ratio from 0.98±0.17 in healthy condition to 0.49±0.24 ($P = 0.02$) on d28, declining together with the advancement of fibrosis (Fig. 5b). Given the different types of sulfation in the total HS composition, a detailed analysis of N- (NS), 2- and 6-sulfation pattern was performed by HPLC (Fig. 5c and Supplementary Fig. 9). Indeed, a marked increase was seen in 6-sulfation (6S) HS portion (Fig. 5c), which is usually associated with a pronounced angiogenic growth[46]. Specifically, 6S HS increased from 2.94±1.11% to 11.87±6.71% ($P = 0.05$) on d7 and to 13.50±5.12% ($P = 0.008$) on d28 post-NSTEMI (Fig. 5c and Supplementary Fig. 9). In addition, NS HS increased from 4.82±0.61% to 8.57±1.93% ($P = 0.15$) on d7 and to 11.88±2.63% ($P = 0.002$) on d28 post-NSTEMI (Fig. 5c and Supplementary Fig. 9).

Despite this observed increase in 6S HS, possibly implying angiogenesis, post-ischaemic fibrotic remodelling compromises physiological vascularity, which is required for normal cardiomyocyte function in healthy conditions. Therefore, to clarify this point in our model of NSTEMI, we looked at the binding capacity of the extracted sulfated HS to angiogenic growth factors, such as VEGF. Binding assays showed a significant drop in the binding capacity 28 days after the surgical procedure ($P = 0.036$) (Fig. 5d), confirming the absence of functional vascularity. In conclusion, ECM GAGs analysis supported the previously observed functional and histological alterations following NSTEMI.

## Discussion

Currently, NSTEMI is the most common presentation of acute MI as most cases with an acute coronary event are NSTEMI patients[4,47–49].

This is partly the result of a widespread use of risk-factor modifying drugs, powerful lipid-lowering statins, and anti-platelet agents (aspirin)[50]. NSTEMI patients have lower in-patient (during their admission for the primary NSTEMI) and short-term mortality rates, but significantly higher long-term mortality than those of STEMI patients[6–8]. A Danish registry study of 8,889 patients showed that the 5-year mortality after NSTEMI was 16%[51], and another registry study highlighted a 10-year survival rate of only around 50%[9]. Nonetheless, to the best of our knowledge, there are currently no large animal models that can reproduce both the functional and histological characteristics of NSTEMIs as a preclinical foundation to study interventions that might ameliorate short and long-term effects of NSTEMI. The animal models currently employed usually adopt the ligation of the LAD at different points, and/or including diagonal branches of either the LAD or the left circumflex artery which would necessarily produce STEMI and transmural infarcts[10,11]. However, without an appropriate preclinical model, interventions based on a traditional full-thickness infarction might lead to clinically ambiguous outcomes[5,52].

Therefore, we have presented a model of NSTEMI that is triggered by multiple ligations (2 cm apart) from the level of the first diagonal artery and parallel to the LAD to within 3–4 cm of the apex and we compared this procedure with animals with full ligation of the first diagonal branch to induce STEMIs. As a result, following NSTEMIs we observed patchy and non-transmural infarcts, as confirmed by ECG and histological changes in the anterolateral wall of the left ventricle. The fourth universal definition of MI requires a rise and fall of cardiac troponin (cTn) and one other criterion from the following: symptoms of acute ischaemia, new ischaemic ECG changes, new Q-waves, loss of viable myocardium or a new wall-motion abnormality in a pattern consistent with ischaemic aetiology via imaging[31,53]. Thus, we have demonstrated that this multiple suture ligation approach results in infarcts that fulfil all of the criteria for an NSTEMI by comparing its functional response with STEMIs. We have observed a significant rise and fall of cTn over time following NSTEMI and STEMI. However, the absolute peak value of cTn does not consistently correlate with the type or size of infarction in NSTEMI; no specific cTn level differentiates STEMI from NSTEMI, but cTn values may be used for risk stratification for early intervention[54]. In addition, typical changes of NSTEMI on ECG in sheep reflected the standard range of changes classified as NSTEMI in the clinical setting. Moreover, histology showed definitive partial thickness myocardial necrosis and fibrosis.

Transthoracic echocardiography (TTE) is a non-invasive and well-recognised tool and the protocol employed in this study is based on validated methods[55]. Nonetheless, several important points should be noted in trying to correlate an animal model of NSTEMI with features of NSTEMI seen in the clinics. Besides the biological parameters used to determine the response of the infarcted, peri-infarcted and unaffected areas after NSTEMI, functional parameters were also employed to correlate these findings. In addition to EF, which is the most widely used parameter to determine and report left ventricular function, FS was also used employing left ventricular EDD and ESD. These factors are also potential markers of left ventricular dilatation in response to myocardial injury. However, left ventricular EDD and ESD were not

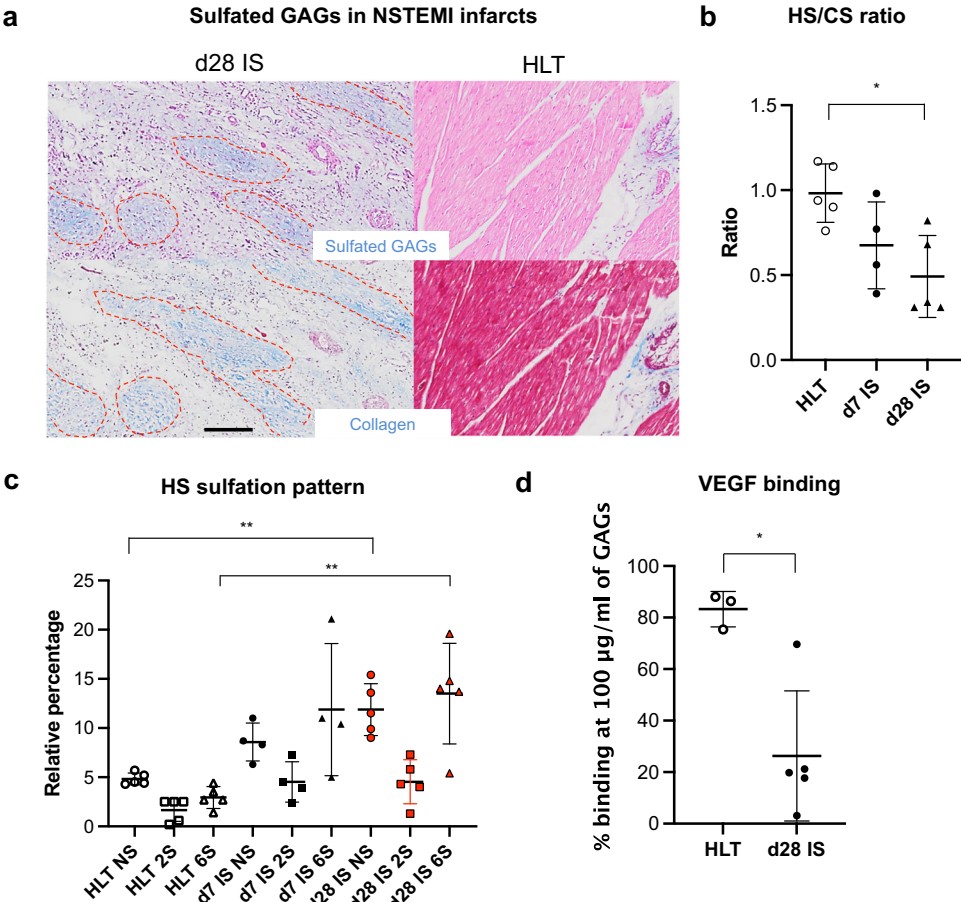

**Fig. 5 | An irreversibly altered extracellular matrix shows specific changes in the HS sulfation pattern following NSTEMI. a** Representative Alcian Blue (top) and Masson's Trichrome (bottom) showing sulfated glycosaminoglycans (GAGs) and collagen (dashed in red) in ischaemic core (IS) region of NSTEMI infarcts on d28 post-NSTEMI. $n = 5$ animals per group. **b** Quantification of heparan sulfate (HS) to chondroitin sulfate (CS) ratio in GAGs extracted from tissue. $n = 5$ HLT, $n = 4$ d7, $n = 5$ d28 animals. **c** Relative percentage of NS, 2S and 6S sulfation in extracted HS from healthy (HLT) and IS samples on d7 and d28 post-NSTEMI. $n = 5$ HLT,
$n = 4$ d7 and $n = 5$ d28 animals. **d** VEGF binding capacity of extracted total GAGs (at 100 µg/ml) between HLT and d28 IS. $n = 3$ HLT and $n = 5$ d28 animals. Kruskal-Wallis with Dunn's multiple comparisons test in (**b**, $P = 0.02$ HLT vs. d28 IS) and (**c**, $P = 0.002$ HLT NS vs. d28 IS NS, $P = 0.008$ HLT 6S vs. d28 IS 6S), two-tailed Mann–Whitney test ($P = 0.036$) in (**d**). *$P < 0.05$, **$P < 0.01$. Data are plotted showing the mean and standard deviation in (**b**–**d**). Source data are provided as a Source Data file.

primary endpoints since the study endpoint was on day 28 post-NSTEMI, which may not have allowed sufficient time for dilation in response to the myocardial injury. Therefore, regional WMI analysis was also evaluated, focusing on the anterolateral walls on TTE. Here we need to consider that scoring is based on a 3-point scoring system indicated by the American Society of Echocardiography and the European Association of Cardiovascular Imaging[56] and remain aware that the difference between a normally contracting wall (1), a hypokinetic wall (2), and even an akinetic wall (3) can be subtle in some cases and may vary depending on the observer. Two cardiologists reviewed all the echocardiograms independently to reduce potential observer error and, in case of disagreement, there was a discussion and consensus on the scoring.

The current model reflects rising clinical presentation in NSTEMI patients who develop myocardial injury and reduction in EF with a significant risk of heart failure in the long-term. Specifically, the average decrease in EF after the surgical procedure is lower than most of the reductions seen after the total occlusion of the LAD[10,11,27,28]. This was also confirmed within our cohort of sheep subjected to STEMI. The variability in the decrease of this functional parameter reflects the range in clinical cases where multiple factors can influence the functional outcome[22]. A main limitation of this model would be the open thoracotomy nature to create infarcts. Moreover, although the infarcts

are non-transmural, there is a mixture of subendocardial to epicardial infarcts. Epicardial infarcts may impact wall tension differently in the long-term compared with subendocardial infarcts, resulting in different outcomes in left ventricular geometry. Nonetheless, histological, gene expression and protein analyses indicated that the damaged areas followed the same irreversible fibrotic pattern seen using STEMI ovine models[20,28]. In addition, extended effects of cardiomyocyte necrosis in the border zone were seen at the mitochondrial level, including the accumulation of dense bodies inside mitochondria, as previously reported after intracoronary balloon occlusion in a swine model[57].

Post-ischaemic remodelling is tightly coupled with profound alterations in the organisation and composition of the cardiac ECM[33,36]. By focusing on the molecular changes in the transcriptome and proteome of the presented NSTEMI model, the relevance of ECM components such as glycoproteins, which were only partially known within this context in previous studies[18,58], has clearly emerged in this study. Here, we derived from pathway analyses on gene expression data the importance of molecular changes in glycans occurring during the post-ischaemic remodelling (day 7 and 28), rather than performing a steady-state characterisation of the cardiac ECM[18,58]. This allowed us to associate specific glycan structural changes (NeuGc, NeuAc, α-(2,3)- and α-(2,6)-linkage-type) with the timing of the post-ischaemic

inflammatory phase. Indeed, recent technical advances in processing and identifying glycans by mass spectrometry have been used as tools to elucidate their biological role[59–61]. These glycan structure profiles are associated with the inflammatory cascade initiated by the need to clear dead cardiomyocytes by infiltrating immune cells. As shown by data on Siglecs, the increased presence of such ligands for sialylated moieties can be likely ascribed to the massive recruitment of monocytes – further differentiating in macrophages – in the harsh ischaemic microenvironment during the inflammatory phase. Further studies are warranted to elucidate how the recruitment of inflammatory cell populations can be modulated towards beneficial remodelling, for instance to enhance cardiac muscle repair by switching macrophage polarisation from a destructive to an anti-inflammatory effect.

Specifically, LC-ESI-MS/MS on N-linked glycans confirmed the previously reported increase in sialylation[58] and identified crucial changes such as the marked abundance of NeuGc and terminal α-gal only at the inflammatory phase (day 7). Moreover, since most studies evaluating the relevance of NeuGc relate either to cancer biology or to immunotherapy pertaining to xenogeneic reactions[62–64], data reported in this study constitute a key finding on the presence of NeuGc in myocardial tissue. A relevant study observed a higher NeuGc/NeuAc ratio in adult than in neonatal myocardial tissue[65]. However, this increase in NeuGc was only tenuously associated with cardiomyocyte development, and the precise extent of this change was not defined[65]. In relation to terminal α-gal increase seen in the proposed model on day 7 post-NSTEMI, so far an increased expression of this marker was demonstrated in extensive studies carried out by the Galili group in wound healing models[66,67]. In addition, α-gal had been found specifically on N-linked glycans in bovine, equine and porcine pericardium[68]. Thus, the increased abundance of α-gal in the inflammatory phase post-MI may be mainly due to the temporary recruitment of a disordered and highly proliferative granulation-type tissue.

Triple gene knockout (GGTA1/CMAH/β4GalNT2, TKO) pig[69,70], in which the expression of α-gal and NeuGc is eliminated, is likely to be an optimal source of organs for transplantation. The TKO pig's tissue is normal compared to wild type pig[71], indicating that these xeno-auto-antigens are not crucial for biological heart function. When the ovine model was used to produce biotherapeutics or to evaluate them against carbohydrate antigens (CA), the xeno-auto-antigens would be a problematic barrier. Here, the model is employed to reveal the rationales of NSTEMI. Indeed, both epitopes are highly regulated by the immune system created by NSTEMI, which is in line with changes of sialylation upon inflammation from MI patients' serum[43]. Therefore, by dissecting the putative N- and O-glycan structures present in the cellular membrane and ECM fractions, we have identified distinct changes in the glycoprofile pattern through cardiac post-ischaemic remodelling.

Alterations in glycan expression were also present in ECM components such as GAGs which, unlike the changes in cellular membrane N- and O-glycan composition reported above, contain well-known glycan moieties. Importantly, GAGs are among the structurally fundamental moieties of the ECM; they also play a relevant role in both the toll-like receptor (TLR)-related inflammatory response and the enhancement of angiogenesis through the binding to growth factors[72–74]. We defined how the balance between HS and CS varied post-ischaemia further to advance the current knowledge on cardiac post-ischaemic remodelling. Thus, after quantifying the different GAGs across the remodelling time points and by focusing on HS sulfate pattern, we identified increases in markers of angiogenesis on day 7 post-NSTEMI (6S HS). Despite the well-known interaction of 6S HS with angiogenic growth factors[46], VEGF binding assay data excluded an actual sustained angiogenic effect throughout post-ischaemic remodelling. Therefore, the current model of NSTEMI would include the timely formation of a non-functional vasculature bed in the affected ischaemic area, as it also occurs after STEMI induction in large animals[75].

Overall, this study defines an ovine NSTEMI model resembling clinical non-transmural infarcts, which is the most prominent type among hospitalised patients. Following validation of functional differences compared to STEMIs, pathway analyses based on extensive omics (transcriptomics and proteomics) paved the way to identify alterations in glycan profiles over post-ischaemic remodelling. Specific glycan structures—also shared in human data—underscored the involvement of Siglecs in the recruited inflammatory cells within the ischaemic core and border zone regions. Further studies would be needed to design therapeutic strategies to modulate the glycan pattern we identified in the cellular membrane and ECM, thereby alleviating the long-term prognosis of this type of infarction.

## Methods
### NSTEMI ovine model
**Ethical approval and animal welfare.** The in vivo experiments were carried out in a licensed laboratory in the Faculty of Veterinary Medicine, Lithuanian University of Health Sciences (LSMU), in accordance with European Directive (2010/63/EU), under the Permission Nr. G2-59. The experimental protocol was approved by the Commission on Ethics of Experimental Animals under the State Food and Veterinary Service of the Republic of Lithuania. Ethics approval and oversight were also provided by the Faculty of Veterinary Medicine, LSMU. A veterinary team performed all surgical procedures and provided post-operative animal care. Romanov eight- to ten-month-old adult male sheep (35 kg weight on average) were housed in an experimental animal facility certified for animal housing and welfare. Three to four animals shared a dwelling space (1.5 m² per animal) with straw for bedding and a 12/12 (day/night) light cycle. Sheep were fed only with hay *ad libitum*, and no concentrated food was given so as not to gain additional weight.

**Premedication and surgical procedure.** To induce either NSTEMI or STEMI, sheep were first sedated with Xylazine (1 mg/kg intramuscular (IM), Bela–Pharm GmbH & Co.KG, Germany) as premedication, then moved to the medication table and injected with Butorphanol (0.2 mg/kg IM, Richter Pharma, Austria) before shaving. Hair was removed on the left side of the chest, sternum, jugular vein and spine of the pelvic areas and animals were transferred into the preparation room. Under sterile conditions venous catheters (16 G, Provein™, Lars Medicare, India) were inserted into the left jugular vein. Saline solution was delivered intravenously at a rate of 1 ml/kg/h. Ketamine (7 mg/kg intravenous (IV)) was used to maintain anaesthesia and sheep were prepared for intubation. The larynx was visualised through a laryngoscope and the mouth cavity was sprayed with 10% lidocaine spray solution (Egis Pharmaceuticals PLC, Hungary) to avoid respiratory spasm. The endotracheal tube (ET, Kruuse, Denmark) was secured, the cuff inflated and artificial lung ventilation performed with a Large Animal Volume Controlled Ventilator (model 613, Harvard apparatus, USA). Long-term deep anaesthesia was obtained using Propofol (5–6 mg/kg/9 min IV, Fresenius Kabi, Germany) and Sevoflurane 2% (Baxter, Belgium) by inhalation (oxygen was regulated around 21% to keep the animal adequately oxygenated during anaesthesia). A nasopharyngeal probe (DeRoyal Industries Inc., USA) was passed into the nasopharynx for core temperature monitoring. Constant temperature of 38 °C was maintained on the heated top surgery table. Sheep were carefully monitored using both manual practices and mechanical tools: eye position, palpebral reflex, mucous membrane colour and capillary refill time were checked and recorded at five-minute intervals. Vital monitoring was performed through ECG, pulse oximetry, temperature, and oscillometric blood pressure monitoring. Specifically, intermittent oscillometric blood pressure monitoring at five-minute intervals, continuous pulse oximetry and nasopharyngeal monitoring were performed and recorded using a Draeger Vista 120 monitor (Shanghai Draeger Medical Instrument Co. Ltd, China). ECG was employed to monitor NSTEMI and STEMI induction throughout the

procedure and duration of anaesthesia using a VE-300 (Gima, Italy) ECG machine. Magnesium (2 mg/kg, Pfizer, USA), and amiodarone (1.5 mg/kg, Pfizer, USA) were intravenously injected in sheep before the surgical procedure to induce MI. Following MI, amiodarone (0.01 mg/kg/min, IV infusion) was administered for 1 h to prevent ventricular arrhythmias. During and following surgeries, animals received benzylpenicillin three times a day (600 mg, IM, Pfizer, USA), and streptomycin twice daily (500 mg, IM, Pfizer, USA), for five days. Flunixin meglumine (2.2 mg/kg, IV, Excella GmBH, Germany) was used as analgesic for five days. Whenever lung oedema was detected, hydrocortisone (250 mg, IV, Pfizer, USA) was administered three times a day.

Induction of NSTEMI was performed by multiple strategic coronary artery ligations on the left ventricle lateral to and parallel to the LAD. Specifically, a left lateral thoracotomy was performed through the fourth intercostal space, followed by a pericardiotomy. Deep non-transmural ligations were performed with 2/0 Prolene® (J&J Ethicon EMEA, Belgium) at 2 cm intervals lateral and parallel to the LAD from the level of the first diagonal moving distally towards and up to 3–4 cm from the apex (Supplementary Fig. 1). Care was taken not to ligate the first diagonal branch. Seven additional sheep were used to induce STEMIs by full-occlusion of the first LAD diagonal branch. The blue Prolene® sutures used to ligate the coronaries were cut long enough to allow for the identification and tracking of the infarction site after hearts were explanted. The pericardium was closed with 4/0 Prolene® after obtaining absolute haemostasis to limit post-operative adhesions. A chest tube was placed with its tip in the pericardial sac and the remnant holes in the left chest before the thoracotomy was closed in layers, and the animal recovered. The animals were given analgesia and fluids post-operatively as outlined by the institutional protocol.

**Functional analyses post-surgery.** Echocardiography measurements were recorded the day before each surgical procedure and at the study time points of 7 and 28 days. All echocardiographic examinations were performed in calm, unsedated standing animals. A 5 MHz probe was employed and the console and software used were Mindray DP-7 (Mindray Medical, China). Images and windows for the echocardiographic protocol were derived from techniques described for horses and more recently adapted for sheep[55]. Two cardiologists performed examinations and agreed on the interpretation and derivation of the data. Six two-dimensional (2-D) parasternal images were obtained from the right, and three 2-D parasternal images from the left. Indices captured were EF, FS, LV volumes and diameters as well as regional wall motion. Blood samples were drawn for serial serum troponin measurements at baseline and following STEMI/NSTEMI induction at day 1, 2 and 3 and sent to a local hospital laboratory for analysis.

**Heart explantation, sampling and histology**
At the endpoints of the study (day 7 and day 28 post-NSTEMI), sheep were anaesthetised as detailed in the description of the surgical induction of NSTEMI. After reopening the thoracotomy wound and pericardium, hearts were explanted. After heart explantation, animals were euthanised using Exagon (40 mg/kg, IV, Richter Pharma AG, Austria). Perfusion with phosphate buffer saline (PBS) was performed twice to wash out any remaining blood following the explantation. Each explanted heart was axially sectioned keeping a thickness of 1 cm for each slice. NSTEMI infarcts were visible by whitish colouring of the affected regions in the left ventricle and confirmed by the blue polypropylene ligating sutures. Explant images were taken with a Cybershot DSC-HX200V camera (Sony, Japan). Tissue harvesting was performed by taking multiple samples (0.5 cm maximum) from the ischaemic site, border and remote area. Tissue processing considered the optimal conditions according to the future analysis to be performed[76,77]. Specifically, samples for histology were submerged in

4% paraformaldehyde (PFA) O/N at 4 °C. Histological analyses were performed using samples embedded in optimal cutting temperature compound (OCT) medium (Sakura Finetek, USA), and either stained with Alcian Blue solution at pH 2.5 (Sigma-Aldrich, USA) or Masson's Trichrome staining (Sigma-Aldrich, USA). For immunofluorescence staining, samples were blocked with 10% donkey serum (Sigma-Aldrich, USA) for 1 h at room temperature (RT) and then incubated overnight (O/N) at 4 °C with rabbit polyclonal antibody to α-SMA (ab5694, Abcam, UK) at 1:200 dilution or rabbit polyclonal antibody to collagen type I (ab34710, Abcam, UK) at 1:200 dilution. Samples were then washed with TBS-T (tris-buffered saline, Tween 20 0.05%) and incubated with anti-rabbit Alexa Fluor® 647-donkey secondary antibody at 1:500 dilution in TBS-T for 1 h at RT. After the incubation, samples were washed with TBS-T, mounted with Prolong® Gold with DAPI (Thermo Fisher Scientific, USA), and coverslipped. Glasses were left to cure overnight at RT before sealing. A FluoViewTM 1000 confocal microscope (Olympus, Japan) was used for all imaging and images analysed using ImageJ v.1.51 (National Institutes of Health, USA).

**Electron microscopy analysis**
On d7 and d28 post-NSTEMI-induction tissue samples from healthy and ischaemic, border and remote infarcted hearts were fixed in 2.5% glutaraldehyde at 4 °C for 24 h. Tissue processing for transmission electron microscopy analysis (TEM) started by washing samples in phosphate buffer (PB) and post-fixing in 1% osmium tetraoxide (Sigma Aldrich, USA) for 2 h at RT. Following this, samples were gradually dehydrated in 30%, 50%, 70%, 95% and pure ethanol. Then, two washes in pure acetone were performed before the embedding. Samples to be embedded were first infiltrated with Araldite® epoxy resin (Electron Microscopy Sciences, USA) at a progressive 2:1, 1:2 ratio with acetone and finally with pure resin for 24 h each at RT. To conclude the processing, samples were moved to freshly-made pure Araldite® epoxy resin, and then samples were left to cross-link at 60 °C for 48 h. Once embedded, ultrathin sections (70-90 nm thick) were cut into samples using a diamond glass knife and were then put onto copper grids. Tissue sections were post-stained with 0.5% uranyl acetate (Laboratory Instruments & Supplies, Ireland) for 3 min and 3% lead citrate (Laboratory Instruments & Supplies, Ireland) for 5 min using an EM AC20 Auto Ultrastainer® (Leica, Germany).

**RNA sequencing**
Following tissue harvesting, samples (0.5 cm maximum) from the ischaemic site, border and remote areas of infarcted or from left ventricular wall healthy hearts were harvested and stored in RNAlater™ (Thermo Fisher Scientific, USA) at −80 °C until further processing. Each tissue sample (40 mg) was finely cut with a scalpel, digested in TRIzol™ (Thermo Fisher Scientific, USA) by bead grinding using a TissueLyser LT (Qiagen, Germany) and extracted by the phenol-chloroform method. Specifically, once samples were completely homogenised, RNA was extracted using RNeasy® Mini Spin Columns (Qiagen, Germany), as per manufacturer's instructions.

RNA sample yield, quality and integrity were assessed by Qubit® 2.0 Fluorometer (Life Technologies, USA) and Agilent TapeStation (Agilent Technologies, USA). RNA library preparation, sequencing and bioinformatics analysis were conducted at GENEWIZ, Inc. (USA). Briefly, NEBNext® Ultra™ RNA Library Prep Kit (New England Biolabs, USA) was used for RNA-seq library preparation. Agilent TapeStation (Agilent Technologies, USA) was employed to validate the sequency library and cDNA were quantified using Qubit® 2.0 Fluorometer (Invitrogen, USA) as well as by quantitative PCR (KAPA Biosystems, USA). Sequencing libraries were loaded on the Illumina® HiSeq 4000 (Illumina Inc., USA) in high output mode and samples sequenced using a 2 × 150 paired end configuration. The HiSeq Control Software conducted image analysis and base calling. Raw sequence data (.bcl files) generated from Illumina® HiSeq was converted into fastq files and

de-multiplexed using Illumina's bcl2fastq 2.17 software. One mis-match was allowed for index sequence identification. Sequence reads were trimmed to exclude adapter sequences and poor quality nucleotides using Trimmomatic v.0.36. Reads mapping was performed using STAR aligner v.2.5.2b on the human reference genome available on ENSEMBL. Unique gene hit counts were calculated using feature Counts from the Subread package v.1.5.2 and counted only when falling within exon regions. Following gene hit count extraction, data were used to perform differential expression gene (DEG) analysis. Comparison of gene expression between the groups of samples was performed within the package DESeq2.

### Gene ontology enrichment
Database for Annotation, Visualization, and Integrated Discovery (DAVID, http://david.ncifcrf.gov) was used to perform Gene ontology (GO) enrichment analysis on DEG data from RNA-seq. We obtained the data plots by combining expression data with functional analysis in the R package GOplot (http://cran.r-project.org/web/packages/GOplot). Biological processes identified functional classes of genes and biological processes GO terms were queried. Significant terms were determined as those with $P < 0.05$. The top ten biological processes for each time point were chosen after removing processes with greater than 80% redundancy.

### Quantification of heparan to chondroitin sulfate ratio
At the time of tissue harvesting around 200 mg of left ventricular tissue from healthy and peri-infarct areas on d7 and d28 post-NSTEMI were snap-frozen to be processed to extract total sulfated GAGs. First, dried-powdered samples were weighed and suspended in a buffer to a final concentration of 25 mg of tissue/ml. Tissue digestion was performed by incubating the tissue suspension with proteinase K (PK, 50 µg/ml, Sigma-Aldrich, USA) at 56 °C O/N. After enzymatic inactivation at 90 °C for 30 min, Dnase (7.5 U/ml, Qiagen, Germany) was added and samples were incubated O/N. Lipid elimination was performed by chloroform extraction[45]. After GAGs dialysis, 1,9-dimethylmethylene blue (DMMB) assay was used to quantify GAGs. HS and CS quantities were determined by incubating digested samples with a cocktail of heparinases[78] (Iduron, UK). Specifically, chondroitinase ABC (25 mU/sample, 2 h at 37 °C) was used for specific CS elimination. The absence of a significant abundance of other GAGs in tissue samples was checked by combining both heparinases and chondroitinases.

### Release of N- and O-linked glycans from membrane proteins
Snap-frozen healthy and infarcted (40 mg from ischaemic core, border and remote regions) were digested using the Mem-PER™ Plus Membrane Protein Extraction Kit (Thermo Fisher Scientific, USA) to extract cytosolic and membrane protein fractions as per manufacturer's instructions. Protein concentration was determined using a Micro BCA™ Protein Assay Kit (Thermo Fisher Scientific, USA). To extract N-linked glycans from membrane protein fractions, ischaemic, border and remote region samples were derived from three different animals per group were used. Protein extraction buffer was exchanged to 7 M urea through a 30 kDa MWCO centrifugal filter (Millipore, USA), followed by incubation with 25 mM dithiothreitol (DTT) at 56 °C for 45 min. After reduction, samples were alkylated with 62.5 mM iodoacetamide (IAA) at RT for 50 min in the dark. Then, samples were trypsinised (sequencing grade 1% w/w, Promega, USA) O/N at 37 °C and resulting peptides were precipitated with 80% (v/v) acetone. Pellets were left to dry, then washed twice with cold 60% methanol, and air-dried. To release N-linked glycans, samples were incubated with PNGase F (Asparia Glycomics, Spain) in ammonium acetate (50 mM, pH 8.4) O/N at 37 °C. Then, a SEP-Pak C18 cartridge (Waters Corporation, USA) was used to separate N-linked glycans from O-glycopeptides. Specifically, a SEP-Pak C18 cartridge was conditioned with dilutions (90% and 10%) of acetonitrile (ACN) in 0.5%

trifluoroacetic acid (TFA). Once the sample was applied, elution with 5% acetic acid allowed the release of N-linked glycans. Moreover, O-glycopeptides were eluted by adding 65% ACN in 0.5% TFA. After drying at 45 °C, N-linked glycans were reduced by 0.5 M sodium borohydride (NaBH$_4$) and 20 mM NaOH O/N at 50 °C. In addition, reductive β-elimination reaction enabled the release of O-linked glycans by incubation in a buffer containing 0.5 M NaBH$_4$ and 50 mM NaOH O/N at 50 °C. Reaction quenching was performed with glacial acetic acid and then samples were desalted and dried[79].

### LC-ESI-MS/MS glycomic analysis
After following the procedure to release N- and O-linked glycans from membrane protein samples, LC-ESI MS/MS was run to perform the glycomic analysis[80]. A packed in-house column (10 cm × 250 µm) with 5 µm porous graphite particles (Hypercarb™, Thermo Fisher Scientific, USA) was used to separate oligosaccharides. Then, following oligosaccharide injection on to the column, samples were eluted with an acetonitrile (ACN) gradient (Buffer A, 10 mM ammonium bicarbonate; Buffer B, 10 mM ammonium bicarbonate in 80% ACN). A 40 cm × 50 µm i.d. fused silica capillary was used as a transfer line to the ion source. A Linear Trap Quadrupole (LTQ)-mass spectrometer (Thermo Fisher Scientific, USA), with an IonMax standard ESI source was used to analyse samples in negative ion mode. The heated capillary was kept at 270 °C, and the capillary voltage was −50 kV. Each sample was analysed by full scan ($m/z$ 380-2000 two microscans, maximum 100 ms, target value of 30,000), together with MS$^2$ scans (two microscans, maximum 100 ms, target value of 10,000) with normalised collision energy of 35%, isolation window of 1.0 units, activation q = 0.25 and activation time 30 ms). The threshold for MS$^2$ was set to 300 counts. Xcalibur™ v.2.0.7 (Thermo Fisher Scientific, USA) was used for data acquisition and processing. The putative glycan structures were identified in the samples by manual annotation from their MS/MS spectra. Importantly, assumptions were made to indicate structural annotations. Briefly, N- and O-linked glycan structures were assumed to follow the classic biosynthetic pathways. In addition, diagnostic fragmentation ions to determine N- and O-glycans was performed[60]. To identify α-linked Gal, terminal hexose-hexose units were considered.

Following MIRAGE guidelines[81] for glycomic analysis, N- and O-glycan annotated structures were submitted and are currently available at the Unicarb-DB database link https://unicarb-dr.glycosmos.org/references/352. To compare relative N- and O-glycan abundance across different samples, each structure was quantified relative to the total content by integration of the extracted ion chromatogram peak area. Specifically, the Area Under the Curve (AUC) of the N- / O-glycan structure was normalised to the total AUC and indicated as a percentage. Analysis of the peak area was processed using Progenesis QI (Waters Corporation, USA). Clustering analysis was conducted using Clustering Explorer v.3.5 (HCE 3.5, University of Maryland, http://www.cs.umd.edu/hcil/hce/hce3.html).

### Protein extraction and digestion for proteomic analysis
Ischaemic, border and remote region samples (30–50 mg) from NSTEMI-infarcted sheep were snap-frozen following heart explantation. On the day of tissue processing, samples were thawed and finely cut into small pieces with a scalpel before starting tissue digestion and protein extraction. RIPA lysis and extraction buffer (Thermo Fisher Scientific, USA) with cOmplete™ EDTA-free protease inhibitor cocktail (Roche, Switzerland) was used to resuspend the samples before homogenisation (2 cycles 15 min each) using a TissueLyser LT (Qiagen, Germany) set at 50 oscillations per min. Once completely homogenised, samples were incubated on ice for 15 min and centrifuged at 12,000 rpm for 10 min at 4 °C. Total protein concentration was determined by Micro BCA™ Protein Assay Kit (Thermo Fisher Scientific, USA) after setting a titration curve. Then, protein samples were concentrated and detergents removed by a suspension trapping (S-Trap) method[82].

Specifically, S-trap™ Micro spin columns (ProtiFi, USA) were used following the manufacturer's instructions, with minor modifications. Briefly, Tris-HCl was used instead of TEAB (Triethylammonium bicarbonate) buffer. Approximately 100 µg of protein sample was dried and resuspended for further reduction (20 mM DTT, 10 min 95 °C) and alkylation (40 mM IAA, 30 min in the dark). After incubation in aqueous phosphoric acid at 1.2% final concentration, binding buffer was added and samples were loaded into micro-columns for protein trapping and trypsinisation (1 h at 37 °C), as per manufacturer's protocol. Peptide elution was centrifuged at 4000 g with ammonium bicarbonate (50 mM) and 0,2% formic acid (FA). Recovery of hydophobic peptides was performed with 50% ACN with 0.2% FA. The final peptide concentration was determined using a NanoDrop™ One/Onec Microvolume UV–Vis Spectrophotometer (Thermo Fisher Scientific, USA).

### nLC-ESI MS/MS label-free proteomic analysis

After digestion and sample preparation, approximately 1.5 µg of peptide per sample was injected into nano Ultra-High-Performance Liquid Chromatography (UHPLC) system (Ultimate™ 3000 RSLCnano, Thermo Fisher Scientific, USA) coupled online with Impact HD™ UHR-QqToF (Bruker Daltonics, Germany). Sample analysis was performed twice to reduce variability. Sample loading was conducted using first a pre-column (Dionex, Acclaim PepMap 100 C18, cartridge, 300 µm), and then a 50 cm nano-column (Dionex, ID 0.075 mm, Acclaim PepMap100, C18). Sample separation occurred at 40 °C with a flow rate of 300 nL/min using multistep 4 h gradients of ACN[82]. The column was connected to a nanoBoosterCaptiveSpray™ ESI source (Bruker Daltonics, Germany). Collision-Induced Dissociation (CID) fragmentation ($N_2$ as collision gas) was applied as data-dependent acquisition mode. Before each sample was run, a specific lock mass (1221.9906 m/z) and a calibration segment (10 mM sodium formate cluster solution) were applied to improve mass accuracy[83]. DataAnalysis™ v.4.1 Sp4 (Bruker Daltonics, Germany) was used to elaborate data and protein identities and relative abundancies were determined using Peaks Studio 8.5 (Bioinformatics Solutions Inc., USA)[84]. Each sample was run and analysed as two independent replicates. For protein identification, Uniprot's reference database of *Ovis aries* was accessed on Feb 2018, 556,825 sequences; 199,652,254 residues. In particular, the following parameters were set: enzymatic digestion performed by trypsin, allowing one missed cleavage; precursor mass tolerance was 20 ppm; fragment mass tolerance of 0.05 Da; carbamidomethylation as fixed modification. False-positive identification rate (FDR) was set ≤1%, and a peptide score of $-Log(p\text{-value}) \geq 20$ was considered adequate for confident protein identification. De Novo ALC score was set ≥50%. Relative peptide signal intensity was calculated only for confidently identified peptide features. Then, the AUC of each extracted ion chromatogram was calculated and used for the relative quantification after normalisation Total Ion Current (TIC). Cumulative peak areas of the proteins were measured by considering only unique peptides assigned to specific proteins. Only proteins with more than two unique peptides were considered for the analysis.

### Ingenuity Pathway Analysis (IPA®) analysis of transcriptomic and proteomic data

DEG data were processed as.xls files. The gene list was sorted to set all the data relative to the different groups to a summed-up ID gene list. The created.xls file had columns reporting each condition's $\log_2$(fold-change) DEG data compared to the healthy animals. As per the manufacturer's instructions, this file was loaded in IPA® (Qiagen, Germany) software. Here, $\log_2$(fold-change) > 1.5 and < −1.5 and adjusted-$P$ < 0.05 were applied as cut-offs Diseases&Functions, Canonical Pathway and Causal Network Analyses[85]. Comparative analysis was conducted on the files of core ischaemic samples on days 7 and 28 post-NSTEMI. Canonical pathway and upstream regulators data were compared by their activation z-score and associated $P$. Identified protein data from nLC-ESI-MS/MS label-

free quantification with a unique peptide number above 2 were selected and listed in a.xls file. Protein entries from the infarcted samples were reported in terms of fold-change ratio to healthy conditions. The.xls file was loaded in the IPA® (Qiagen, Germany) software according to the instructions. Cut-offs of Fold change >1.5 and <−1.5 were applied for Upstream Regulator Analysis.

### Electrocardiograms data from NSTEMI patients

The electrocardiograms (ECGs) which validated functional changes consistent with induced NSTEMI in sheep (Supplementary Fig. 3b) were standard 12-lead ECGs in human subjects. These patients presented with signs and symptoms suggestive of an MI which were subsequently confirmed on the ECG and serum troponin measurements at University Hospital Galway (Ireland). The protocol for obtaining consistent ECGs is based on hospital protocols derived from American Heart Association (AHA) recommendations[86]. Neither Institutional Review Board approval nor informed consent was required for the use of electrocardiogram data from NSTEMI patients as it falls under the category of Non-Human Subject Research, as the research team did not have access to identifiers or keys to link coded data (even temporarily). They are part of archived teaching vignettes for undergraduate and postgraduate students.

### Statistics and reproducibility

The in vivo study was designed and powered for statistical significance (power = 0.8) to determine the sample size. Animals were randomly selected for each experimental group and investigators were blinded to allocation during experiments. All animals were named and given an identification number in the chronological order of surgery-induction using a systematic labelling for identification. Animals from at least two different batches of surgeries were analysed for each group in all analyses to avoid a single batch-dependent bias. No data were excluded from the analyses. Statistical analyses were performed using Prism® v9.0.0 (GraphPad Software Inc., USA) and differences with $P < 0.05$ were considered significant. Normality and equality of variance were tested before a statistical test. Multiple group analyses were performed by running either Kruskal–Wallis or Tukey's one-way ANOVA. For RNA-seq analysis, two-sided Wald test using Benjamini-Hochberg method was used to generate $p$-values and $\log_2$(fold-change) as per DEseq2 method. Adjusted p-value calculations to identify the most significant results in IPA® Analyses were based on the Benjamini–Hochberg method of accounting for multiple testing. Statistical differences were defined as $*P < 0.05$, $**P < 0.01$, $***P < 0.001$.

### Reporting summary

Further information on research design is available in the Nature Portfolio Reporting Summary linked to this article.

## Data availability

The authors declare that all data supporting the findings of this study are available from the authors on request. RNA-seq raw data used in this study are available in GEO database under accession code GSE164245 at the link https://www.ncbi.nlm.nih.gov/geo/query/acc.cgi?acc=GSE164245. Proteomic data were deposited in MassIVE repository (accession code MSV000089576) and are available at https://massive.ucsd.edu/ProteoSAFe/dataset.jsp?task=2bf4eed7d3484de9b99759cff4011ccf. Glycomics-derived annotated structures were deposited in the Unicarb-DB database and are available at the link https://unicarb-dr.glycosmos.org/references/352. Source data are provided with this paper.

## Code availability

No custom computer code or mathematical algorithm that is deemed central to the conclusions was used in this study.

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

## Acknowledgements

This study was funded by the European Commission funding under the AngioMatTrain 7th Framework Programme (Grant Agreement Number 317304) and by the research grant from Science Foundation Ireland (SFI) co-funded under the European Regional Development Fund under Grant Number 13/RC/2073 and13/RC/2073_P2 to A.P. The authors acknowledge the use of the Centre for Microscopy and Imaging facilities at the University of Galway. Mass spectroscopy analysis of glycans was performed by the Swedish infrastructure for biological mass spectrometry (BioMS) supported by the Swedish Research Council. Prof. P. Dockery, P. Lalor and E. McDermott from the Anatomy Department at the University of Galway assisted in processing for TEM analysis. R. Grigalevičiūtė and A. Kučinskas from the LSMU assisted with veterinary procedures and in ensuring animal well-being. Prof. D.H. Pauza and Dr K. Rysevaite from the Institute of Anatomy at the LSMU assisted with initial processing of tissue samples. The authors would like to acknowledge Dr R. Bohara for editorial assistance, M. Doczyk for assistance in drawing the schematics and Dr O. Carroll for technical help. This manuscript is dedicated to the memory of Renato Sartorello.

## Author contributions

P.C., M.D.C. and A.P. conceived the idea and designed the experiments. M.D.C., V.V., M.R. and P.C. developed and validated the ovine model of MI and performed the in vivo study. A.G.P. assisted with the in vivo experiments. P.C., R.S. and F.F. performed the analyses from in vivo experiments. A.K. and E.E. assisted in functional data recording and analysis. V.Z. coordinated all veterinary procedures and animal welfare. P.C., R.S. and F.F. ran analyses on gene expression and proteomic data. C.J. and N.G.K. performed the LC-ESI-MS/MS glycomic analysis. C.C. and F.M. performed the nLC-ESI-MS/MS proteomic analysis. S.C. and D-P.G. performed the GAGs analysis. C.V. revised the statistical analyses. A. C. critically revised the clinical data of the manuscript and assisted in revision. P.C. wrote the manuscript. M.D.C. and R.S. edited the manuscript. A.P. supervised the entire project.

## Competing interests

The authors declare no competing interests.

## Additional information

¹CÚRAM, SFI Research Centre for Medical Devices, University of Galway, Galway, Ireland. ²Department of Molecular Medicine, University of Padova, Padova, Italy. ³LSMU Biological Research Center, Lithuanian University of Health Sciences, Kaunas, Lithuania. ⁴Department of Cardiology, Medical Academy, Lithuanian University of Health Sciences, Kaunas, Lithuania. ⁵Institute of Cardiology, Lithuanian University of Health Sciences, Kaunas, Lithuania. ⁶Proteomics Core Facility at Sahlgrenska Academy, University of Gothenburg, Gothenburg, Sweden. ⁷Glycobiology, Cell Growth and Tissue Repair Research Unit (Gly-CRRET), University Paris Est Créteil, Créteil, France. ⁸Clinical Proteomics and Metabolomics Unit, School of Medicine and Surgery, University of Milano-Bicocca, Vedano al Lambro, Italy. ⁹Institute of Anatomy, Lithuanian University of Health Sciences, Kaunas, Lithuania. ¹⁰Manaaki Manawa Centre for Heart Research, Department of Physiology, Faculty of Medical & Health Sciences, University of Auckland, Auckland, New Zealand. ¹¹UF Health Heart and Vascular Hospital, Gainesville, FL, USA. ¹²Department of Cardiac, Thoracic, Vascular Sciences and Public Health, University of Padova, Padova, Italy. ¹³Section of Pharmacy, Department of Life Sciences and Health, Faculty of Health Sciences, Oslo Metropolitan University, Oslo, Norway. ¹⁴These authors contributed equally: Paolo Contessotto, Renza Spelat. ✉e-mail: abhay.pandit@universityofgalway.ie; mark.dacosta@universityofgalway.ie

