## [Peer Review File · Nature Communications]

Reproducing extracellular matrix adverse remodelling of non-ST myocardial infarction in a large animal modelReviewers' Comments:

Reviewer #1:

Remarks to the Author:

Understanding mechanistic and molecular mechanisms of ischemic injury to the myocardium has attributed to the development of large animal models close to the human. Sheep resemble the coronary anatomy of humans closely, and the relatively comparable body size of sheep and swine and similar coronary anatomy and vasomotor responsiveness to human make them relevant for utilization of multiple diagnostic and therapeutic strategies. The most popular and classic ovine model of ischemia can be produced by ligating the different coronary vessels. These highly reproducible models require a thoracotomy to occlude the target vessels. In this study the authors ligated the myocardium at different places parallel to the left anterior descending artery without actually blocking the coronary artery itself. By doing so, they tried to mimic non ST elevation myocardial infarction.

I have a number of comments for the authors:

1. did the authors compare their model with one where the left anterior descending coronary artery is completely occluded
2. The ventricular size and ejection fraction do not change much from day 7 to day 14 meaning that not much adverse remodeling is taking place during that time. If extracellular matrix expansion plays an important role in the setting of NSTEMI, then one would expect an increase in the size of the ventricles and worsening LV function
3. Did the investigators measure any hemodynamic parameters in these animals.
4. It is not clear how distinctive patterns in the expression of 51 complex N-glycans and glycosaminoglycans in cellular membranes and ECM improve our understanding of NSTEMI mechanisms especially since they don't affect ventricular dimensions
5. This is a difficult model to establish in large animals but I am not convinced there is any new information that is surprising in this new model. There have been multiple myocardial infarction models in sheep by occluding completely or temporarily the coronary arteries and even though the model described by the authors is different in that the myocardial ischemic damage is less, the proteomic analysis does not advance our understanding of the mechanisms in NSTEMI. In addition there are no interventions to modulate the effects and there is no comparison to a model with complete occlusion of the coronary artery.

Reviewer #2:

Remarks to the Author:

This manuscript presents very interesting results. In particular, the connection between myocardial infarction and glycans was discovered for the first time in the study. It should therefore be considered for publication. There, however, are things needed to be discussed as mentioned below:

1. It should discuss the reason why the glycan structural profiles have changed and should specify increased and newly produced glycan structures.
2. It should also discuss whether the lectin expression levels of siglecs are increased or not. For example, in Fig. 4, glycans such as 1112.17, 1185.17 and 1185.08 show increased values of sialylated glycans. I would study, for example, the lectins expression levels for siglecs as a partner molecule.
3. The manuscript studies NeuGc and α -galactose present in sheep, but these two structures don't exist in humans. What structures are applied in case of humans?

A point-by-point response to Reviewers' comments:

Reviewer #1

Understanding mechanistic and molecular mechanisms of ischemic injury to the myocardium has attributed to the development of large animal models close to the human. Sheep resemble the coronary anatomy of humans closely, and the relatively comparable body size of sheep and swine and similar coronary anatomy and vasomotor responsiveness to human make them relevant for utilization of multiple diagnostic and therapeutic strategies. The most popular and classic ovine model of ischemia can be produced by ligating the different coronary vessels. These highly reproducible models require a thoracotomy to occlude the target vessels. In this study the authors ligated the myocardium at different places parallel to the left anterior descending artery without actually blocking the coronary artery itself. By doing so, they tried to mimic non ST elevation myocardial infarction.

Response: We thank the Reviewer for the detailed and up-to-date introduction to our study, and the acknowledgement of the relevance and importance of large animal models, in particular ovine models of MI, in the investigation of underlying pathophysiological mechanisms of ischaemic injury.

1. did the authors compare their model with one where the left anterior descending coronary artery is completely occluded.

Response: We acknowledge the relevance of the Reviewer's comment and performed several analyses to compare our current partial-thickness (NSTEMI) model with a widely employed model of coronary artery ligation to produce transmural (STEMI) infarcts. As requested, we have provided additional data from sheep having the same age and gender, which were subjected to ligation of the first LAD diagonal to create STEMI in the same and comparable territory. Functional (EF), serological (troponin I), hemodynamic (stroke volume and cardiac output), and ECGs data were combined within the new **Extended Data Fig 2** for the Reviewer to make a clear comparison between NSTEMI and STEMI in sheep. As expected, a clear difference in functional and serological parameters resulted from STEMIs compared to NSTEMIs.

The title of the first **results section** has now been modified to "*Functional impairment in a model of NSTEMI compared to STEMI*", and the text revised on:

Page 4, lines 130-134: "The proposed model of NSTEMI was performed in a cohort of 21 sheep and compared with five additional sheep subjected to full-occlusion of the first LAD diagonal branch."

Page 5, lines 143-148: "Therefore, to validate the current model, we compared it with a standard full-occlusion ligation of the first diagonal branch which caused full-thickness STEMIs in sheep of same age and gender. By comparing NSTEMI with STEMI, in the latter, we noticed a greater reduction in EF both at d7 ($15.4 \pm 4.4\%$, $p=0.03$) and d28 ($14.8 \pm 3.3\%$, $p=0.03$) post-ligation (Extended Data Fig. 2a), together with a marked rise ($p<0.05$) in troponin I level from d1 to d3 post-surgery (Extended Data Fig. 2b)."

Extended Data Figure 2 | Comparison between STEMI and NSTEMI model

a, Left, ejection fraction (EF) absolute values before ligation (baseline), 7 (d7) and 28 (d28) days after full-occlusion ligation. Right, comparison of the relative decrease in EF on d7 and d28 between STEMI and NSTEMI. ($n=5$ per group) Animals had the same age and weight. **b**, Left, mean troponin I level from d1 to d3 following either STEMI or NSTEMI induction. $n=5$ STEMI and $n=4$ NSTEMI animals. Right, individual troponin I level from ligation to d7 post-surgery in STEMI and NSTEMI. $n=5$ STEMI and $n=4$ NSTEMI animals. **c**, Hemodynamics of

STEMI and NSTEMI in sheep measured as stroke volume (left) and cardiac output (right). $n=5$ STEMI and $n=3$ NSTEMI animals. **d**, Representative electrocardiogram (ECG) before STEMI-induction (left) and post-ligation (right). Marked ST elevation in leads I, II and aVL, reciprocal ST depression in lead aVR and milder ST depression in lead III are circled in red. $n=5$ animals. Data in **a-c** are reported as dot-plots, Kruskal-Wallis test in (**a**, left, and **c**) one-way ANOVA with Tukey's post hoc correction in (**a**, right), Mann-Whitney test in (**b**). $*p<0.05$.

Given these substantial changes, we needed to revise also **Extended Data Fig. 4** to avoid repetitions in presenting troponin I data and the relative text:

Page 6, lines 171-177:

“Importantly, we have confirmed the described NSTEMI induction approach with clinical data since electrocardiograms (ECGs) post-ligation highlighted comparable changes in T wave inversion in leads I, II, III and aVF (Fig. 1e and Extended Data Fig. 3). In line with the current findings on LVEDD and LVESD, the reduction in fractional shortening (FS) on d7 and d28 post-surgery was not significant (Extended Data Fig. 4a). Echocardiography, troponin and ECG data followed the clinical criteria which concur to define the current model as representative of an NSTEMI event³¹.”

Extended Data Figure 4 | Functional and histological evaluation of the NSTEMI model

a, Fractional shortening (FS) percentage at baseline, 7 and 28 days post-NSTEMI. Reduction in FS (delta) on days 7 and 28 after surgery. $n=11$ animals. **b**, Schematics of tissue harvesting from explanted hearts. After perfusion with PBS to remove the excess blood, each heart was cross-sectioned in slices with a thickness of 1 cm from the atrium to the apex. The whitish colour within samples identified clear MI areas with a maximum size of 0.5 cm in every direction were taken from the core ischaemic, the border and the remote regions. Data in **a** is reported as dot-plots, one-way ANOVA with Tukey's post hoc correction, and Wilcoxon test for delta FS.

In addition, we have also revised the Abstract, Introduction and Discussion sections to include the relevance of this comparison between NSTEMI and STEMI throughout the manuscript, as reported below.

Abstract

“Upon histological and functional investigation to validate the proposed model and comparison with STEMI full-ligation model, RNA-seq and proteomics analyses showed the distinctive features of post-NSTEMI tissue remodelling. Transcriptome and proteome-derived pathway analyses at acute (7 days) and late (28 days) post-NSTEMI pinpointed specific alterations in cardiac post-ischaemic extracellular matrix (ECM).”

Introduction

Page 3, lines 91-92: “Therefore, there is a need in the field to adopt clinically relevant models to study NSTEMI pathophysiology and also reveal its functional differences with STEMI induction.”

Page 4, lines 106-113: “Large animals, specifically sheep, have been extensively used to evaluate the recovery of heart functionality following STEMI because of the similarity in organ volume to humans^{20,21}. Therefore, in this study, we first validated the functional differences between NSTEMI and STEMI within an ovine model and further analysed the distinctive molecular features of the NSTEMI model both at an acute (7 days) and a late (28 days) timepoint. Specifically, we have studied the ischaemic, border zone and remote regions at the different timepoints post-NSTEMI by histology, RNA-sequencing, proteomics and glycomics.”

Discussion

Page 12, lines 393-409: “Therefore, we have presented a model of NSTEMI that is triggered by multiple ligations (2 cm-apart) from the level of the first diagonal artery and parallel to the LAD to within 3-4 cm of the apex. Care was taken not to ligate the first diagonal itself. We also compared this procedure with animals which had full-ligation of the first diagonal branch to induce STEMI. As a result, following NSTEMIs’ we observed patchy and non-transmural infarcts, as confirmed by ECG and histological changes in the anterolateral wall of the left ventricle. The 4th universal definition of MI requires a rise and fall of cTn and one other criterion from the following: symptoms of acute ischaemia, new ischaemic ECG Changes, new Q-waves, loss of viable myocardium or a new wall motion abnormality in a pattern consistent with ischaemic aetiology via imaging^{31,53}. Thus, we have demonstrated that this multiple suture-ligation approach results in infarcts that fulfil all of the criteria for an NSTEMI Type 1 infarct by comparing its functional response with STEMI. We have observed a significant rise and fall of cTn over time following NSTEMI and STEMI. However, the absolute peak value of cTn does not consistently correlate with the type or size of infarction in NSTEMI; no specific cTn level differentiates STEMI from NSTEMI, but cTn values may be used for risk stratification for early intervention⁵⁴. In addition, typical changes of NSTEMI on ECG in sheep reflected the standard range of changes classified as NSTEMI in the clinical setting.”

Page 13, lines 435-439: “The current model reflects rising clinical presentation in NSTEMI patients who develop myocardial injury and reduction in EF with the significant risk of heart failure in the long-term. Specifically, the average decrease in EF after the surgical procedure is lower than most of the reductions seen after the total occlusion of the LAD^{10,11,27,28}. This was also confirmed within our cohort of sheep subjected to STEMI.”

2. The ventricular size and ejection fraction do not change much from day 7 to day 14 meaning that not much adverse remodeling is taking place during that time. If extracellular matrix expansion plays an important role in the setting of NSTEMI, then one would expect an increase in the size of the ventricles and worsening LV function.

Response: We understand the point made by the Reviewer. It is true that ventricular impairment (EF) does not change from day 7 to day 28 (typo d14 by the Reviewer), as we stated in the original manuscript and have confirmed in the revised version on *Page 5, lines 151-160*:

“Post-ischaemic remodelling involves different degrees of dilatation, hypertrophy and collagen scarring. This process occurs over weeks and months, and it is influenced by multiple factors, including the size and site of the infarct, whether the infarct was transmural (STEMI) or not (NSTEMI), the amount of stunning of the peri-infarct myocardium, the patency of the related coronary artery and local trophic factors^{29,30}. Since NSTEMI is not the result of complete occlusion of a coronary artery, it usually affects a small area or those that are diffuse or patchy areas of the ventricular muscle rather than the entire thickness of the local ventricular wall. Indeed, as expected, given the nature of this type of MI, the presented model did not significantly vary in left ventricular diastolic and systolic diameters (LVEDD and LVESD) (Fig. 1c).”

Indeed, we do not claim that post-NSTEMI ECM adverse remodeling affects geometrical changes further than initial permanent damage. In essence, *NSTEMI differs from STEMI since there is a focal permanent muscle injury which does not involve the full thickness of the ventricular wall. However, this does not mean that there is a substantial difference in the biological post-ischaemic remodelling process.* In addition, as shown by new data presented in **Extended Data Fig. 2a**, STEMI’s do not necessarily result in a progressive worsening in EF from d7 to d28. Nonetheless, as shown by data comparing EF and troponin I between the two models, in NSTEMI’s (partial-thickness), the overall left ventricular damage is lower and less severe than that observed in STEMI’s (full-thickness).

Extended Data Figure 2 | Comparison between STEMI and NSTEMI model

a, Left, ejection fraction (EF) absolute values before ligation (baseline), 7 (d7) and 28 (d28) days after full-occlusion ligation. Right, comparison of the relative decrease in EF on d7 and d28 between STEMI and NSTEMI. ($n=5$ per group) Animals had the same age and weight. Data in **a-c** are reported as dot-plots, Kruskal-Wallis test in (**a**, left) one-way ANOVA with Tukey’s post hoc correction in (**a**, right).

In addition, in the discussion (Page 13, lines 421-433), we have underscored this model's limitation in assessing ventricular dilation by LVEDD and LVESD already at day 28 post-surgery. However, the wall motion index highlighted apparent ventricular abnormalities.

“FS employs left ventricular end-diastolic and end-systolic diameters (LVEDD and LVESD). These factors are also potential markers of left ventricular dilatation in response to myocardial injury. However, LVEDD and LVESD were not primary endpoints since the study endpoint was on day 28 post-NSTEMI, which may not have allowed sufficient time for dilation in response to the myocardial injury. Therefore, regional wall motion index (WMI) analysis was also utilised in this study, and the focus was on the anterolateral walls on TTE. Here we need to consider that scoring is based on a well-validated 3-point scoring system from the American Heart Association⁵⁶ and remain aware that the difference between a normally contracting wall (1), a hypokinetic wall (2), and even an akinetic wall (3) can be subtle in some cases and may vary depending on the observer. We had two different cardiologists review all the echocardiograms independently to reduce potential observer error. In case of disagreement, there was a discussion and consensus on the scoring.”

3. Did the investigators measure any hemodynamic parameters in these animals.

Response: We acknowledge the relevance of the Reviewer’s comment and added hemodynamic data comparing NSTEMIs with STEMI in **Extended Data Fig. 2c**. Stroke volume and cardiac output measurements were taken at d0, d7 and d28 post-NSTEMI/STEMI and did not show any significant difference between the two models. We have also added a representative ECG to show the typical features of STEMI in sheep subjected to full-occlusion ligation.

The Results section in the main text has been revised on Page 5, lines 148-150:

“ECGs showed typical changes of STEMI. Nonetheless, hemodynamic parameters showed no significant difference in stroke volume and cardiac output (Extended Data Fig. 2c,d).”

Extended Data Figure 2 | Comparison between STEMI and NSTEMI model

c, Hemodynamics of STEMI and NSTEMI in sheep measured as stroke volume (left) and cardiac output (right). $n=5$ STEMI and $n=3$ NSTEMI animals. Data in **c** is reported as dot-plots, Kruskal-Wallis test. **d**, Representative electrocardiogram (ECG) before STEMI-induction (left) and

post-ligation (right). Marked ST elevation in leads I, II and aVL, reciprocal ST depression in lead aVR and milder ST depression in lead III are circled in red. $n=5$ animals.

4. It is not clear how distinctive patterns in the expression of 51 complex N-glycans and glycosaminoglycans in cellular membranes and ECM improve our understanding of NSTEMI mechanisms especially since they don't affect ventricular dimensions.

Response: We understand the relevance of this point made by the Reviewer and revised the manuscript to increase its clarity. NSTEMI functional analyses reflected a smaller - although irreversible – impairment when compared to STEMI (new data provided in Extended Data Fig.2). That said, our glycomics (51 complex N-glycans which were detected) and GAGs data only reflect the biological nature behind the permanent adverse remodelling which constitute the basis for adverse tissue remodelling which is typical of both STEMI and NSTEMI.

Thus, we were not correlating the N-glycan profile with a progressive worsening of geometrical parameters. However, we acknowledge more clearly the limitations of this model in this sense in our revised discussion on *Pages 13-14, lines 442-447*:

“Moreover, although the infarcts are non-transmural, there is a mixture of subendocardial to epicardial infarcts. Epicardial infarcts may impact wall tension differently in the long-term compared with subendocardial infarcts, resulting in different outcomes in left ventricular geometry. Nonetheless, histological, gene expression, and protein analyses indicated that the damaged areas followed the same irreversible fibrotic pattern seen using STEMI ovine models^{20,28}.”

To address the Reviewer’s concerns about the N-glycan distinctive pattern in NSTEMI, in the revised manuscript, we have analysed which specific N-glycan structures in our dataset were shared with those found as increased in a recent study from MI patients (including NSTEMIs). The relative **results** section and **Fig. 4f** have been revised on *Page 10, lines 322-325*:

“To highlight the similarity of the proposed model to the clinical cases of NSTEMI, we report a group of highly present complex sialylated N-glycan structures (m/z 1111.52 (NeuAc₂Hex₅HexNAc₄), 1185.08 (NeuAc₂Hex₅HexNAc₄dHex₁) that are shared (Fig. 4g) with recently characterised ones in MI patients⁴³.”

Figure 4 | Distinct glycoprofile in the infarcted heart following NSTEMI

f, Extracted ion chromatography (EIC) showing the most abundant N-linked glycans in the membrane protein extracts from HLT myocardium and IS at d7 and d28 post-NSTEMI. m/z values framed in red indicate N-glycan structures, which were found to increase also in MI patients' sera by Lim et al.⁴³. Regions of infarcted hearts are labelled as follows: IS= core ischaemic, BZ = border zone, F = remote zone from the infarct. Data are representative of two independent experiments. In **a-f**, each analysed sample was a pool of samples collected from three individuals per group and region and was investigated by PG-LC-ESI-MS/MS.

We have also added specific data analysis in Fig. 4g from RNA-seq to define further the Siglecs ligands present in immune cells, which can bind to the increased sialylated N-glycans found at d7 and d28 post-NSTEMI. Indeed, the increase in NeuGc and terminal alpha-Gal we observed at d7 is associated with granulation tissue development, which does not define a chronic fibrotic condition but a transient early healing response. Furthermore, our findings in the HS sulfate pattern identified increases in markers of non-functional angiogenesis, which does not improve the outcome.

Overall, this study reports for the first time such changes in N-glycome in the specific condition of NSTEMI, thus providing the readers with pioneering biological and molecular data on the actual relevance of such moieties in the confined and specific conditions of this pathology, without associating these findings with a geometrical wall impairment.

Figure 4 | Distinct glycoprofile in the infarcted heart following NSTEMI

g, Gene expression levels of Siglec-1, -2, -10, -11 and -15 at d7 and 28 post-NSTEMI from RNA-seq data on IS and BZ samples. $n=4$ animals per group. Regions of infarcted hearts are labelled as follows: IS= core ischaemic, BZ = border zone. Data are representative of two independent experiments. In (**g**), RNA-seq data analysis was run using DESeq2 (threshold set to $\text{Log}_2(\text{fold change}) > 1.5$), and Wald's test assessed significant differences. * $p < 0.05$, *** $p < 0.001$.

5. This is a difficult model to establish in large animals but I am not convinced there is any new information that is surprising in this new model.

Response: We regret that the Reviewer did not find enough novelty in the original version of the manuscript. Nonetheless, we believe that our revised version - which **includes a functional comparison with STEMIs and additional data on glycomics** - shows a clear distinction of the current model supported by a thorough panel of omics, histology, and functional analyses, as already strongly supported by Reviewer #2.

We have summarised the main aim and results of this manuscript at the end of the discussion on *Page 16, lines 516-525*:

“Overall, this study defines an ovine NSTEMI model resembling clinical non-transmural infarcts that are the most prominent type among hospitalised patients. Following validation of functional differences compared to STEMIs, pathway analyses based on extensive omics (RNA-seq and proteomics) paved the way to identify alterations in glycan profiles over post-ischaemic remodelling. Specific glycan structures – also shared in human data – underscored the involvement of Siglecs in the recruited inflammatory cells within the ischaemic core and border zone regions. Further studies would be needed to design therapeutic strategies to modulate the glycan pattern we identified in the cellular membrane and ECM, thereby alleviating the long-term prognosis of this type of infarction.”

Moreover, we would like to underscore the critical glycomics results now reported in the relevant results section on *Page 10, lines 309-325*:

“By analysing sialic acid linkage-type, we noticed a pattern of peaked increase in α -(2,6)-sialic acid linkage type at d7 in ischaemic regions, and thus within the inflammatory phase, in contrast with a progressive increase of α -(2,3)-sialylation in all regions - infarcted and not - over the remodelling from d7 to d28 (Extended Data Fig. 8b). These findings led to the evaluation of possible association with sialic acids binding ligands which are well known to be expressed by the infiltrating immune cell populations, such as monocytes and macrophages³⁹⁻⁴². Indeed, we have observed that both Siglec-1 and -15 - which are highly present in monocytes and macrophages^{40,41} - were markedly overexpressed ($p < 0.05$) at d7 and significantly dropped 21 days later, in line with a resolved inflammation (Fig. 4g). In addition, Siglec-15 expression pattern was significant both in the ischaemic core ($p < 0.05$) and border

zone ($p < 0.001$) region, and Siglec-10, another marker of infiltrating activated monocytes and macrophages⁴², decreased ($p < 0.001$) from d7 to d28 (Fig. 4g).

To highlight the similarity of the proposed model to the clinical cases of NSTEMI, we report a group of highly present complex sialylated N-glycan structures (m/z 1111.52 (NeuAc₂Hex₅HexNAc₄), 1185.08 (NeuAc₂Hex₅HexNAc₄Hex₁) that are shared (Fig. 4g) with recently characterised ones in MI patients⁴³”

There have been multiple myocardial infarction models in sheep by occluding completely or temporarily the coronary arteries and even though the model described by the authors is different in that the myocardial ischemic damage is less, the proteomic analysis does not advance our understanding of the mechanisms in NSTEMI.

Response: As correctly pointed out by the Reviewer, there are multiple MI models in large animals produced by occlusion of various major coronary arteries. However, all these models produce STEMI infarcts and not NSTEMI infarcts which is the crux of our present. We acknowledge that the degree of myocardial damage is lower than in a typical STEMI infarct, which parallels what is seen in humans, and the new data provided supports this within our study (Extended Data Fig.2). However, the primary purpose of this study was also to comprehensively profile NSTEMI using omics (including proteomics) for the first time. Indeed, we have revised our data to highlight further specific molecular changes by glycomic analysis (Fig. 4g), which emerged only following an initial pathway analysis derived from both RNA-seq (Extended Data Fig.7a) and proteomic data (Fig. 4b). Therefore, it was not possible to decouple the relevance of data based on proteomics from the final discoveries (N-glycans) which were made throughout the study. Moreover, this model will be used for multiple omics data hunting (transcriptome, glycomics, and proteomics), adopting different conditions to decipher target genes/proteins.

Moreover, to clarify this link in our manuscript, we revised the relevant results section *on Page 8, lines 250-260*:

“Importantly, gene-annotation enrichment analysis using Database for Annotation, Visualization, and Integrated Discovery (DAVID) software on RNA-seq data highlighted distinctive glycan alterations in the post-ischaemic remodelling in the current NSTEMI model (Extended Data Fig. 7). Given the outcome of RNA-seq and proteomic pathway analyses, we noticed a key involvement of glycan moieties, since both on d7 N-glycan biosynthesis (Extended Data Fig. 7a) and on d28 glycosaminoglycans (GAGs) biosynthesis (Extended Data Fig. 7b) emerged among the biological categories (KEGG pathways) with the highest enrichment score. As the relevance of glycoproteins in the pathophysiology of MI has just started to be investigated^{18,37}, we have performed advanced glycomics on N-linked glycans extracted from the cellular membrane and ECM proteins in ischaemic, border and remote regions.”

In addition there are no interventions to modulate the effects and there is no comparison to a model with complete occlusion of the coronary artery.

Response: We added additional *in vivo* analyses to compare STEMIs with NSTEMIs. We refer to our reply to comment #1 since it overlaps with this one regarding the lack of data from complete occlusion. Regarding the possible therapeutic strategies mentioned by the

Reviewer, we would like to highlight that the main aim of this study, as reflected by the title of the manuscript, is mainly focused on the NSTEMI pathology itself without proposing a treatment at this point. Indeed, we decided to target a specific gap in the cardiovascular field by presenting a novel model of NSTEMI together with its thorough characterisation without stating any intention also to target it. However, we revised our discussion to include the possibility of exploiting the molecular targets which emerged from the presented data and could be an object of further studies in the field on *Page 14, lines 459-471*:

“This allowed us to associate specific glycan structural changes (NeuGc, NeuAc, α -(2,3)- and α -(2,6)-linkage-type) with the timing of the post-ischaemic inflammatory phase. Indeed, recent technical advances in processing and identifying glycans by mass spectrometry have been used as tools to elucidate their biological role⁵⁹⁻⁶¹. These glycan structure profiles are associated with the inflammatory cascade initiated by the need to clear dead cardiomyocytes by infiltrating immune cells. Indeed, as shown by data on Siglecs expression, the increased presence of such ligands (Siglecs) for sialylated moieties can be likely ascribed to the massive recruitment of monocytes – further differentiating in macrophages – in the harsh ischaemic microenvironment during the inflammatory phase. Further studies in the field would be needed to modulate the recruitment of such inflammatory cell populations toward a beneficial remodelling, for instance, to enhance cardiac muscle repair by switching macrophage polarisation from a destructive to an anti-inflammatory effect.”

Reviewer #2

This manuscript presents very interesting results. In particular, the connection between myocardial infarction and glycans was discovered for the first time in the study. It should therefore be considered for publication. There, however, are things needed to be discussed as mentioned below:

Response: We sincerely thank the Reviewer for the solid support for our study and for pointing out the novelty of connecting MI with glycans for the first time. We have carefully addressed all the comments and suggestions to improve our manuscript further.

1. It should discuss why the glycan structural profiles have changed and specify increased and newly produced glycan structures.

Response: We have carefully considered the Reviewer’s comment and apologise for not pointing out the reasons before.

The **discussion** section has been revised on *Page 14, lines 456-463*:

“Here, we derived from pathway analyses on gene expression data the relevance of molecular changes in glycans occurring during the post-ischaemic remodelling (day 7 and 28), rather than performing a steady-state characterisation of the cardiac ECM^{18,58}. This allowed us to associate specific glycan structural changes (NeuGc, NeuAc, α -(2,3)- and α -(2,6)-linkage type) with the timing of the post-ischaemic inflammatory phase. Indeed, recent technical advances in processing and identifying glycans by mass spectrometry have been used as tools to elucidate their biological role⁵⁹⁻⁶¹.”

Since our model of NSTEMI implicated a clear inflammatory phase due to MI induction, as detected by histology and RNA-seq, the observed alterations in the glycan structural profiles are associated with a highly inflammatory condition. In line with recent findings by Lim et al.

from a cohort of MI patients⁴³ – including also NSTEMI – our data (Extended Data Fig. 7) reflected an initial increase in galactosylated structures (at day 7), followed by a progressive rise in sialylation (from d7 to d28). Therefore, we have revised Fig. 4 to highlight which N-glycan structures at d7 and d28 post-NSTEMI emerged in our model and were also detected in the recently reported human dataset⁴³.

The relevant **results** section and **Fig. 4f** have been revised on *Page 10, lines 322-325*:
 “To highlight the similarity of the proposed model to the clinical cases of NSTEMI, we report a group of highly present complex sialylated N-glycan structures (m/z 1111.52 (NeuAc₂Hex₅HexNAc₄), 1185.08 (NeuAc₂Hex₅HexNAc₄dHex₁) that are shared (Fig. 4g) with recently characterised ones in MI patients⁴³.”

Figure 4 | Distinct glycoprofile in the infarcted heart following NSTEMI

f, Extracted ion chromatography (EIC) showing the most abundant N-linked glycans in the membrane protein extracts from HLT myocardium and IS at d7 and d28 post-NSTEMI. m/z values framed in red indicate N-glycan structures, which were found to increase also in MI patients’ sera by Lim et al.⁴³. Regions of infarcted hearts are labelled as follows: IS= core ischaemic, BZ = border zone, F = remote zone from the infarct. Data are representative of two independent experiments. In **a-f**, each analysed sample was a pool of samples collected from three individuals per group and region and was investigated by PG-LC-ESI-MS/MS.

2. It should also discuss whether the lectin expression levels of siglecs are increased or not. For example, in Fig. 4, glycans such as 1112.17, 1185.17 and 1185.08 show increased values of sialylated glycans. I would study, for example, the lectins expression levels for siglecs as a partner molecule.

Response: We acknowledge the relevance of the comment made by the Reviewer and added new specific analyses in Fig. 4f,g to focus on both the mentioned sialylated glycans and Siglecs expression. RNA-seq data reported the expression of Siglec-1 (CD169/Sialoadhesin), -10, -11, and -15, which are also present in humans⁴². As discussed in the previous point, the increase in sialylated glycans such as those mentioned by the Reviewer can be coupled with a simultaneous overexpression of Siglecs in the inflammatory cells, which are relevant within this post-ischaemic context. Indeed, we noticed a clear pattern of increased Siglecs expression on day 7 post-NSTEMI compared to day 28 and reported it in the revised main Fig. 4g.

The relative **results** section and **Fig. 4g** have been revised on *Page 10, lines 313-321*:

“These findings stimulated us to evaluate the possible association with sialic acids binding ligands known to be expressed by the infiltrating immune cell populations, such as monocytes and macrophages³⁹⁻⁴². Indeed, we have observed that both Siglec-1 and -15 - which are highly present in monocytes and macrophages^{40,41} - were markedly overexpressed ($p < 0.05$) at d7 and significantly dropped 21 days later, in line with a resolved inflammation (Fig. 4g). In addition, Siglec-15 expression pattern was significant both in the ischaemic core ($p < 0.05$) and border zone ($p < 0.001$) region, and Siglec-10, another marker of infiltrating activated monocytes and macrophages⁴², decreased ($p < 0.001$) from d7 to d28 (Fig. 4g).”

Figure 4 | Distinct glycoprofile in the infarcted heart following NSTEMI

g, Gene expression levels of Siglec-1, -2, -10, -11 and -15 at d7 and 28 post-NSTEMI from RNA-seq data on IS and BZ samples. $n=4$ animals per group. Regions of infarcted hearts are labelled as follows: IS= core ischaemic, BZ = border zone, F = remote zone from the infarct. Data are representative of two independent experiments. In (**g**), RNA-seq data analysis was run using DESeq2 (threshold set to $\text{Log}_2(\text{fold change}) > 1.5$), and Wald’s test assessed significant differences. * $p < 0.05$, *** $p < 0.001$.

The **discussion** section has been revised on *Page 14, lines 463-468*:

“These glycan structures profiles are associated with the inflammatory cascade initiated by the need to clear dead cardiomyocytes by infiltrating immune cells. Indeed, as shown by data on Siglecs expression, the increased expression of such ligands (Siglecs) for sialylated moieties can be likely ascribed to the massive recruitment of monocytes – further differentiating in macrophages – in the harsh ischaemic microenvironment during the inflammatory phase.”

3. The manuscript studies NeuGc and α -galactose present in sheep, but these two structures don’t exist in humans. What structures are applied in case of humans?

Response: We agree that NeuGc and terminal α -galactose are not expressed in humans. NeuGc and alpha-gal (α -gal) are xeno-auto-antigens associated with chronic inflammation

and autoimmune disease in humans. To discuss this point, we have included the following considerations in the revised discussion on Page 15, lines 488-499:

“Triple gene knockout (GGTA1/CMAH/ β 4GalNT2, TKO) pig^{69,70}, in which the expression of α -gal, NeuGc, and Sda is eliminated, are likely to be an optimal source of organs for transplantation. The TKO pig’s tissue is normal compared to WT pig⁷¹ indicating these xeno-auto-antigens are not crucial for biological heart function. When the ovine model was used to produce biotherapeutics or evaluate biotherapeutics against carbohydrate antigens (CA), the xeno-auto-antigens would be a problematic barrier. Here, the model is employed to reveal the rationales of NSTEMI. Indeed, both epitopes are highly regulated by the immune response created by NSTEMI, which is in line with changes of sialylation upon inflammation from MI patients’ serum⁴³. Here, by dissecting the putative N- and O-glycan structures present in the cellular membrane and ECM fractions, we have identified for the first time the precise changes in the glycoprofile pattern through cardiac post-ischaemic remodelling.”

References

1. Roth GA *et al.* Global burden of cardiovascular diseases and risk factors, 1990-2019: Update from the GBD 2019 Study. *J. Am. Coll. Cardiol.* **76**, 2982-3021 (2020)
2. Xie Y, Xu E, Bowe B, Al-Aly Z. Long-term cardiovascular outcomes of COVID-19. *Nat. Med.* Feb 7 (2022)
3. Bahit MC, Kochar A, Granger CB. Post-myocardial infarction heart failure. *JACC Heart Fail.* **6**, 179-186 (2018).
4. McManus DD *et al.* Recent trends in the incidence, treatment, and outcomes of patients with STEMI and NSTEMI. *Am. J. Med.* **124**, 40-47 (2011)
5. Roger VL *et al.* Trends in incidence, severity, and outcome of hospitalized myocardial infarction. *Circulation* **121**, 863-869 (2010)
6. Ishihara M *et al.* Long-term outcomes of non-ST-elevation myocardial infarction without creatine kinase elevation - The J-MINUET Study. *Circ. J.* **81**, 958-965 (2017)
7. Rea F, Ronco R, Pedretti RFE, Merlini L, Corrao G. Better adherence with out-of-hospital healthcare improved long-term prognosis of acute coronary syndromes: Evidence from an Italian real-world investigation. *Int. J. Cardiol.* **318**, 14-20 (2020)
8. Vora AN *et al.* Differences in short- and long-term outcomes among older patients with ST-elevation versus non-ST-elevation myocardial infarction with angiographically proven coronary artery disease. *Circ. Cardiovasc. Qual. Outcomes.* **9**, 513-522 (2016)
9. Erdem G *et al.* Rates and causes of death from non-ST elevation acute coronary syndromes: ten year follow-up of the PRAIS-UK registry. *Int. J. Cardiol.* **168**, 490-494 (2013)
10. Gabisonia K *et al.* MicroRNA therapy stimulates uncontrolled cardiac repair after myocardial infarction in pigs. *Nature* **569**, 418-422 (2019)
11. D’Uva G *et al.* ERBB2 triggers mammalian heart regeneration by promoting cardiomyocyte dedifferentiation and proliferation. *Nat. Cell. Biol.* **17**, 627-638 (2015)
12. Weston C, Reinoga K, van Leeven R, Demian V. Myocardial Ischaemia National Audit Project - How the NHS cares for patients with heart attacks. Annual Public Report April 2014-March 2015 (NICOR Report, University College London, 2017)
13. Moainie SL *et al.* An ovine model of postinfarction dilated cardiomyopathy. *Ann. Thorac. Surg.* **74**, 753-760 (2002)
14. Schmitto JD *et al.* A novel, innovative ovine model of chronic ischemic cardiomyopathy induced by multiple coronary ligations. *Artif. Organs.* **34**, 918-922 (2010)
15. Hashimoto H, Olson EN, Bassel-Duby R. Therapeutic approaches for cardiac regeneration and repair. *Nat. Rev. Cardiol.* **15**, 585-600 (2018)
16. Farbehi N *et al.* Single-cell expression profiling reveals dynamic flux of cardiac stromal, vascular and immune cells in health and injury. *Elife* **26**, 8:e43882 (2019)
17. Tombor LS *et al.* Single cell sequencing reveals endothelial plasticity with transient mesenchymal activation after myocardial infarction. *Nat. Commun.* **12**, 681 (2021)
18. Parker BL *et al.* Quantitative N-linked glycoproteomics of myocardial ischemia and reperfusion injury reveals early remodeling in the extracellular environment. *Mol. Cell. Proteomics.* **10**, M110.006833 (2011)

19. Zhao RR *et al.* Targeting chondroitin sulfate glycosaminoglycans to treat cardiac fibrosis in pathological remodeling. *Circulation*. **137**, 2497-2513 (2018)
20. Ifkovits JL *et al.* Injectable hydrogel properties influence infarct expansion and extent of postinfarction left ventricular remodeling in an ovine model. *Proc. Natl. Acad. Sci. USA*. **107**, 11507-11512 (2010)
21. Macarthur JW Jr *et al.* Preclinical evaluation of the engineered stem cell chemokine stromal cell-derived factor 1 α analog in a translational ovine myocardial infarction model. *Circ. Res.* **114**, 650-659 (2014)
22. Miller AL *et al.* Left ventricular ejection fraction assessment among patients with acute myocardial infarction and its association with hospital quality of care and evidence-based therapy use. *Circ. Cardiovasc. Qual. Outcomes*. **5**, 662-671 (2012)
23. Sugiyama T *et al.* Differential time trends of outcomes and costs of care for acute myocardial infarction hospitalizations by ST elevation and type of intervention in the United States, 2001-2011. *J. Am. Heart Assoc.* **4**, e001445 (2015)
24. Mihalko E, Huang K, Sproul E, Cheng K, Brown AC. Targeted treatment of ischemic and fibrotic complications of myocardial infarction using a dual-delivery microgel therapeutic. *ACS Nano*. **12**, 7826-7837 (2018)
25. Kaul P *et al.* Incidence of heart failure and mortality after acute coronary syndromes. *Am. Heart J.* **165**, 379-85.e2 (2013)
26. Arora S *et al.* Impact of type 2 myocardial infarction (MI) on hospital-level MI outcomes: Implications for quality and public reporting. *J. Am. Heart Assoc.* **7**, e008661 (2018)
27. Dixon JA *et al.* Targeted regional injection of biocomposite microspheres alters post-myocardial infarction remodeling and matrix proteolytic pathways. *Circulation* **124** (2011)
28. Houtgraaf JH *et al.* Intracoronary infusion of allogeneic mesenchymal precursor cells directly after experimental acute myocardial infarction reduces infarct size, abrogates adverse remodeling, and improves cardiac function. *Circ. Res.* **113**, 153-166 (2013)
29. Pfeffer MA, Braunwald E. Ventricular remodeling after myocardial infarction. Experimental observations and clinical implications. *Circulation*. **81**, 1161-1172 (1990)
30. Warren SE, Royal HD, Markis JE, Grossman W, McKay RG. Time course of left ventricular dilation after myocardial infarction: influence of infarct-related artery and success of coronary thrombolysis. *J. Am. Coll. Cardiol.* **11**, 12-19 (1988)
31. Hilliard AL, Winchester DE, Russell TD, Hilliard RD. Myocardial infarction classification and its implications on measures of cardiovascular outcomes, quality, and racial/ethnic disparities. *Clin. Cardiol.* **43**, 1076-1083 (2020)
32. Mühlfeld C, Nyengaard JR, Mayhew TM. A review of state-of-the-art stereology for better quantitative 3D morphology in cardiac research. *Cardiovasc. Pathol.* **19**, 65-82 (2010)
33. Cahill TJ, Choudhury RP, Riley PR. Heart regeneration and repair after myocardial infarction: translational opportunities for novel therapeutics. *Nat. Rev. Drug. Discov.* **16**, 699-717 (2017)
34. Homans DC *et al.* Regional function and perfusion at the lateral border of ischemic myocardium. *Circulation*. **71**, 1038-1047 (1985)
35. Driesen RB *et al.* Structural remodelling of cardiomyocytes in the border zone of infarcted rabbit heart. *Mol. Cell. Biochem.* **302**, 225-232 (2007)
36. Frangogiannis NG. Cardiac fibrosis: Cell biological mechanisms, molecular pathways and therapeutic opportunities. *Mol. Aspects Med.* **65**, 70-99 (2019)
37. Weil BR, Neelamegham S. Selectins and immune cells in acute myocardial infarction and post-infarction ventricular remodeling: Pathophysiology and novel treatments. *Front. Immunol.* **10**, 300 (2019)
38. Contessotto P *et al.* Distinct glycosylation in membrane proteins within neonatal versus adult myocardial tissue. *Matrix Biol.* **85-86**, 173-188 (2020)
39. Affandi AJ *et al.* CD169 Defines activated CD14⁺ monocytes with enhanced CD8⁺ T cell activation capacity. *Front. Immunol.* **12**:697840 (2021)
40. Lerkvaleekul B *et al.* Siglec-1 expression on monocytes is associated with the interferon signature in juvenile dermatomyositis and can predict treatment response. *Rheumatology (Oxford)*. **61**, 2144-2155 (2022)
41. Wang J *et al.* Siglec-15 as an immune suppressor and potential target for normalization cancer immunotherapy. *Nat. Med.* **25**, 656-666 (2019)
42. Angata T, Varki A. Discovery, classification, evolution and diversity of Siglecs. *Mol. Aspects Med.* **18**:101117 (2022)
43. Lim SY *et al.* N-glycan profiles of acute myocardial infarction patients reveal potential biomarkers for diagnosis, severity assessment and treatment monitoring. *Glycobiology*. **32**, 469-482 (2022)
44. Rouet V *et al.* A synthetic glycosaminoglycan mimetic binds vascular endothelial growth factor and modulates angiogenesis. *J. Biol. Chem.* **280**, 32792-32800 (2005)

45. Huynh MB *et al.* Age-related changes in rat myocardium involve altered capacities of glycosaminoglycans to potentiate growth factor functions and heparan sulfate-altered sulfation. *J. Biol. Chem.* **287**, 11363-11373 (2012)
46. Ferreras C *et al.* Endothelial heparan sulfate 6-O-sulfation levels regulate angiogenic responses of endothelial cells to fibroblast growth factor 2 and vascular endothelial growth factor. *J. Biol. Chem.* **287**, 36132-36146 (2012)
47. Alkhouli M *et al.* Age-stratified sex-related differences in the incidence, management, and outcomes of acute myocardial infarction. *Mayo Clin. Proc.* **96**, 332-341 (2021)
48. Mozaffarian D *et al.* Heart disease and stroke statistics--2015 update: a report from the American Heart Association. *Circulation.* **131**, e29-322 (2015)
49. Darling CE *et al.* Survival after hospital discharge for ST-segment elevation and non-ST-segment elevation acute myocardial infarction: a population-based study. *Clin. Epidemiol.* **5**, 229-236 (2013)
50. Weidmann L *et al.* Pre-existing treatment with aspirin or statins influences clinical presentation, infarct size and inflammation in patients with de novo acute coronary syndromes. *Int. J. Cardiol.* **275**, 171-178 (2019)
51. Alzuhairi KS *et al.* Long-term prognosis of patients with non-ST-segment elevation myocardial infarction according to coronary arteries atherosclerosis extent on coronary angiography: a historical cohort study. *BMC Cardiovasc. Disord.* **17**, 279 (2017)
52. Lindsey ML *et al.* Guidelines for experimental models of myocardial ischemia and infarction. *Am. J. Physiol. Heart Circ. Physiol.* **314**, H812-H838 (2018)
53. Thygesen K *et al.* Fourth Universal Definition of Myocardial Infarction (2018). *Circulation.* **138**, e618-e651 (2018)
54. Roffi M *et al.* 2015 ESC Guidelines for the management of acute coronary syndromes in patients presenting without persistent ST-segment elevation: task force for the management of acute coronary syndromes in patients presenting without persistent st-segment elevation of the European Society of Cardiology (ESC). *Eur. Heart J.* **37**, 267-315 (2016)
55. Hallowell GD, Potter TJ, Bowen IM. Reliability of quantitative echocardiography in adult sheep and goats. *BMC Vet. Res.* **8**, 181 (2012)
56. Gottdiener JS *et al.* American Society of Echocardiography recommendations for use of echocardiography in clinical trials. *J. Am. Soc. Echocardiogr.* **17**, 1086-1119 (2004)
57. Galindo CL *et al.* Anti-remodeling and anti-fibrotic effects of the neuregulin-1 β glial growth factor 2 in a large animal model of heart failure. *J. Am. Heart Assoc.* **3**, e000773 (2014)
58. Yang S, Chatterjee S, Cipollo J. The glycoproteomics-MS for studying glycosylation in cardiac hypertrophy and heart failure. *Proteomics Clin Appl.* **12**:e1700075 (2018)
59. Jensen PH, Karlsson NG, Kolarich D, Packer NH. Structural analysis of N- and O-glycans released from glycoproteins. *Nat. Protoc.* **7**, 1299-1310 (2012)
60. Everest-Dass AV, Abrahams JL, Kolarich D, Packer NH, Campbell MP. Structural feature ions for distinguishing N- and O-linked glycan isomers by LC-ESI-IT MS/MS. *J. Am. Soc. Mass Spectrom.* **24**, 895-906 (2013)
61. Lavery SB *et al.* Advances in mass spectrometry driven O-glycoproteomics. *Biochim. Biophys. Acta.* **1850**, 33-42 (2015)
62. He D *et al.* Generation and characterization of a IgG monoclonal antibody specific for GM3 (NeuGc) ganglioside by immunizing β 3Gn-T5 knockout mice. *Sci. Rep.* **8**, 2561 (2018)
63. Hernández AM *et al.* Characterization of the antibody response against NeuGcGM3 ganglioside elicited in non-small cell lung cancer patients immunized with an anti-idiotypic antibody. *J. Immunol.* **181**, 6625-6634 (2008)
64. Barone A, Benktander J, Teneberg S, Breimer ME. Characterization of acid and non-acid glycosphingolipids of porcine heart valve cusps as potential immune targets in biological heart valve grafts. *Xenotransplantation.* **21**, 510-522 (2014)
65. Montpetit ML *et al.* Regulated and aberrant glycosylation modulate cardiac electrical signaling. *Proc. Natl. Acad. Sci. USA.* **106**, 16517-16522 (2009)
66. Wigglesworth KM *et al.* Rapid recruitment and activation of macrophages by anti-Gal/ α -Gal liposome interaction accelerates wound healing. *J. Immunol.* **186**, 4422-4432 (2011)
67. Hurwitz ZM, Ignatz R, Lalikos JF, Galili U. Accelerated porcine wound healing after treatment with α -gal nanoparticles. *Plast Reconstr Surg.* 2012 Feb;129(2):242e-251e
68. Jin C *et al.* Identification by mass spectrometry and immunoblotting of xenogeneic antigens in the N- and O-glycomes of porcine, bovine and equine heart tissues. *Glycoconj. J.* **37**, 485-498 (2020)

69. Zhang R *et al.* Reducing immunoreactivity of porcine bioprosthetic heart valves by genetically-deleting three major glycan antigens, GGTA1/ β 4GalNT2/CMAH. *Acta Biomater.* **72**, 196-205 (2018)
70. Estrada JL *et al.* Evaluation of human and non-human primate antibody binding to pig cells lacking GGTA1/CMAH/ β 4GalNT2 genes. *Xenotransplantation.* **22**, 194-202 (2015)
71. Wang RG *et al.* Antigenicity of tissues and organs from GGTA1/CMAH/ β 4GalNT2 triple gene knockout pigs. *J. Biomed. Res.* **33**, 235-243 (2018)
72. Olivares-Silva F *et al.* Heparan sulfate potentiates leukocyte adhesion on cardiac fibroblast by enhancing Vcam-1 and Icam-1 expression. *Biochim. Biophys. Acta.* **1864**, 831-842 (2018)
73. Taylor KR, Gallo RL. Glycosaminoglycans and their proteoglycans: host-associated molecular patterns for initiation and modulation of inflammation. *FASEB J.* **20**, 9-22 (2006)
74. Prante C *et al.* Transforming growth factor beta1-regulated xylosyltransferase I activity in human cardiac fibroblasts and its impact for myocardial remodeling. *J. Biol. Chem.* **282**, 26441-26449 (2007)
75. Gkontra P *et al.* Deciphering microvascular changes after myocardial infarction through 3D fully automated image analysis. *Sci. Rep.* **8**, 1854 (2018)
76. Zhu W *et al.* Regenerative potential of neonatal porcine hearts. *Circulation.* **138**, 2809-2816 (2018)
77. Barbosa I *et al.* Improved and simple micro assay for sulfated glycosaminoglycans quantification in biological extracts and its use in skin and muscle tissue studies. *Glycobiology.* **13**, 647-653 (2003)
78. Dubail J *et al.* SLC10A7 mutations cause a skeletal dysplasia with amelogenesis imperfecta mediated by GAG biosynthesis defects. *Nat. Commun.* **9**, 3087 (2018)
79. Schulz BL, Packer NH, Karlsson NG. Small-scale analysis of O-linked oligosaccharides from glycoproteins and mucins separated by gel electrophoresis. *Anal. Chem.* **74**, 6088-6097 (2002)
80. Liu Y *et al.* The minimum information required for a glycomics experiment (MIRAGE) project: improving the standards for reporting glycan microarray-based data. *Glycobiology.* **27**, 280-284 (2017)
81. Chinello C *et al.* Proteomics of liquid biopsies: Depicting RCC infiltration into the renal vein by MS analysis of urine and plasma. *J. Proteomics.* **191**, 29-37 (2019)
82. Liu X *et al.* Intraluminal proteome and peptidome of human urinary extracellular vesicles. *Proteomics Clin. Appl.* **9**, 568-573 (2015)
83. Zhang J *et al.* PEAKS DB: de novo sequencing assisted database search for sensitive and accurate peptide identification. *Mol. Cell. Proteomics.* **11**, M111.010587 (2012)

Reviewers' Comments:

Reviewer #1 (Remarks to the Author):

The authors have effectively addressed all the issues I raise with the initial manuscript

Reviewer #2 (Remarks to the Author):

All my comments have been adequately addressed.
I believe that the revised manuscript should be published as it is.
Great work.

NCOMMS-22-19862-A-Z

A point-by-point response to Reviewers' comments:

Reviewer #1

The authors have effectively addressed all the issues I raise with the initial manuscript

Response: We are grateful to the Reviewer for Her/His thoughtful comments.

Reviewer #2

All my comments have been adequately addressed.

I believe that the revised manuscript should be published as it is.

Great work.

Response: We are delighted by the support given by the Reviewer.